# SARS-CoV-2 antibody dynamics and transmission from community-wide serological testing in the Italian municipality of Vo'

Ilaria Dorigatti [1,10✉], Enrico Lavezzo [2,10✉], Laura Manuto[2], Constanze Ciavarella [1], Monia Pacenti[3], Caterina Boldrin[3], Margherita Cattai[3], Francesca Saluzzo [2], Elisa Franchin[2], Claudia Del Vecchio[2], Federico Caldart [2], Gioele Castelli [2], Michele Nicoletti [2], Eleonora Nieddu[2], Elisa Salvadoretti[2], Beatrice Labella[2], Ludovico Fava[2], Simone Guglielmo [2], Mariateresa Fascina[2], Marco Grazioli[2], Gualtiero Alvisi[2], Maria Cristina Vanuzzo[3], Tiziano Zupo[3], Reginetta Calandrin[3], Vittoria Lisi[3], Lucia Rossi[3], Ignazio Castagliuolo[2], Stefano Merigliano [4], H. Juliette T. Unwin [1], Mario Plebani[5], Andrea Padoan[5], Alessandra R. Brazzale[6], Stefano Toppo[2,7], Neil M. Ferguson [1], Christl A. Donnelly [1,8] & Andrea Crisanti[2,3,9✉]

In February and March 2020, two mass swab testing campaigns were conducted in Vo', Italy. In May 2020, we tested 86% of the Vo' population with three immuno-assays detecting antibodies against the spike and nucleocapsid antigens, a neutralisation assay and Polymerase Chain Reaction (PCR). Subjects testing positive to PCR in February/March or a serological assay in May were tested again in November. Here we report on the results of the analysis of the May and November surveys. We estimate a seroprevalence of 3.5% (95% Credible Interval (CrI): 2.8–4.3%) in May. In November, 98.8% (95% Confidence Interval (CI): 93.7–100.0%) of sera which tested positive in May still reacted against at least one antigen; 18.6% (95% CI: 11.0–28.5%) showed an increase of antibody or neutralisation reactivity from May. Analysis of the serostatus of the members of 1,118 households indicates a 26.0% (95% CrI: 17.2–36.9%) Susceptible-Infectious Transmission Probability. Contact tracing had limited impact on epidemic suppression.

[1] MRC Centre for Global Infectious Disease Analysis and the Abdul Latif Jameel Institute for Disease and Emergency Analytics, School of Public Health, Imperial College London, London, UK. [2] Department of Molecular Medicine, University of Padova, Padova, Italy. [3] Azienda Ospedale Padova, Padova, Italy. [4] Department of Surgery, Oncology and Gastroenterology, University of Padova, Padova, Italy. [5] Department of Medicine, University of Padova, Padova, Italy. [6] Department of Statistical Sciences, University of Padova, Padova, Italy. [7] CRIBI Biotech Centre, University of Padova, Padova, Italy. [8] Department of Statistics, University of Oxford, Oxford, UK. [9] Department of Life Science Imperial College London, Exhibition Road, London, UK. [10]These authors contributed equally: Ilaria Dorigatti, Enrico Lavezzo. ✉email: i.dorigatti@imperial.ac.uk; enrico.lavezzo@unipd.it; a.drcrisanti@imperial.ac.uk

In Europe, the successful implementation of non-pharmaceutical interventions (NPIs) in the first wave of SARS-CoV-2 transmission was followed by a resurgence in transmission in the autumn 2020[1,2], requiring the implementation of a new tier-based system in several countries and most recently, stay-at-home orders[3–5]. The development of rapid diagnostic tests[6] and the approval of new, effective vaccines[7–9] provide hope for the future, though fundamental knowledge gaps must be addressed to fully exploit these new tools[10–12]. There are still several uncertainties around the routes and settings where transmission occurs. Within-household transmission has been suggested to play an important role in SARS-CoV-2 transmission based on the reported within-household secondary attack rates[13,14], quantifying the proportion of household members of an infected subject also infected by SARS-CoV-2. However, infection occurs in multiple settings, and it is important to discern the likelihood of acquiring the infection within or outside the household. An accurate measure of the extent of within-household transmission is provided by the probability that a household member acquires the infection from an infectious household member, which can be estimated using mathematical modelling.

The great majority of SARS-CoV-2 infections, irrespective of symptom onset, develop antibodies against different viral antigens[15,16] and mount a significant T-cell mediated response, as documented by recent analyses of the T-cell antigen receptor repertoire[17,18]. This immune response seems to confer some level of protection against re-infection, as supported by the observation that re-infections are rare amongst individuals previously exposed to SARS-CoV-2[19,20], but there are still key questions concerning the duration of immune-mediated protection and its capability to block virus replication upon re-exposure. Over the past year, while waiting for vaccine development and production, substantial resources have been invested in contact tracing aiming to systematically isolate cases and contacts and thereby interrupt transmission chains. This approach has generated disappointing results in Europe and most of the Western world[21], as it proved to be unable to achieve SARS-CoV-2 control. A better understanding of the different routes and settings where infection occurs, along with quantitative estimates of the impact of interventions and of the persistence of the immune response in SARS-CoV-2 exposed individuals can be used to improve both vaccination and non-pharmaceutical interventions.

We show here the results of two sequential serological and oronasopharyngeal swab surveys conducted in the Italian municipality of Vo' in May and November 2020 using three different assays quantifying IgG antibodies targeting the spike (S) and nucleocapsid (N) as well as the total (IgM and IgG) antibodies against the N antigen. These surveys follow two previous oronasopharyngeal swab surveys conducted in the same population in February and March 2020[22]. We estimated the population-level seroprevalence, quantified the magnitude of the antibody response and its persistence, investigated their association with severity, health, and demographic indicators and report on the association between infection and co-morbidities as well as medication history. We also used information on the serological status of 2,566 household members to estimate the Susceptible Infectious Transmission Probability (SITP), which is the probability of SARS-CoV-2 transmission occurring between each susceptible-infectious pair of individuals, measured over the whole period of infectiousness of the infectious individual and in the case when the susceptible individual is not infected by a third party during that period, and is an alternative, more nuanced measure of within-household transmission intensity than the within-household secondary attack rate. We estimated the sensitivity of contact tracing by combining the results of the large-scale PCR surveys conducted in Vo' with the records of the intensive contact tracing efforts implemented at the start of the pandemic by the local health authorities. In a counterfactual analysis, we explore the impact that contact tracing alone, in the absence of the mass testing campaigns and lockdown implemented in February and March 2020[22], would have had on the epidemic dynamics.

## Results

**Investigating serum reactivity to viral antigens**. The May survey involved 2,602 participants, most living in Vo' or in nearby countryside settlements, collectively defined as Vo' cluster (Fig. 1) using three distinct immunological assays detecting antibodies against the S and N antigen (Tables 1 and 2). In total 88.5% of the participants (2,303 subjects) also took part in at least one of the surveys conducted in February and March 2020[22]. Out of the 2,602 tested participants, 162 (6.2%) showed the presence of virus specific antibodies in at least one of the three assays (Fig. 1), and 27.2% of these sera had neutralising antibody titres > 1:40 (1/dil). A total of 2,303 (88.5%) sera did not show the presence of antibodies in any of the three assays (2 had an equivocal DiaSorin result). A further 137 sera (5.3%) were tested only with one of the three assays, because of lack of material, and all of them gave a negative result (1 gave an equivocal DiaSorin result). Out of the 2,602 tested subjects, 2,443 (93.9%) showed a clear reactivity profile to all three assays (i.e., a negative or positive test result, excluding the equivocal DiaSorin results).

**Identification of SARS-CoV-2 exposed individuals**. The precise identification of individuals exposed to the virus has important implications for the assessment of antibody prevalence, evaluation of antibody persistence, correlation with co-morbidities as well as for reconstructing the extent of within-household transmission. We generated three ground truth definitions to identify infected individuals by using increasingly stringent criteria based on PCR results, serum reactivity profile, serum neutralisation titres, and contact history with variable levels of stringency (Supplementary Methods, section 2). Supplementary Table 1 summarises the definitions used, and the number of subjects identified as exposed, according to the different criteria. None of the participants tested positive by PCR in May and we found a single PCR positive subject in November.

**Seroprevalence estimates**. Using a multinomial likelihood (see Methods, likelihood-based seroprevalence estimate) fitted to the observed test results combinations (Table S2) we estimate a prevalence of SARS-CoV-2 infection in Vo' in May of 3.5% (95% Credible Interval (CrI) 2.8–4.3%) and the assay-specific performance results reported in Table 3. This analysis builds on a validation experiment[23] using a group of sera collected one year before the onset of the pandemic (true negatives) as well as sera from individuals with confirmed SARS-CoV-2 infection (true positives) to assess the in-house sensitivity and specificity of the assays (Table 3). The estimated SARS-CoV-2 infection prevalence obtained from the multinomial likelihood (yellow horizontal line and shading in Fig. 2) closely matches the combined assay-specific seroprevalence estimates adjusted by the assay-specific sensitivity and specificity estimated from the model and reported in Table 3 and Fig. 2a. The likelihood-based estimate is completely independent from the ground truth definition; however, it closely matches the seroprevalence estimate obtained using the baseline ground truth definition (Fig. 2b). The assay-specific seroprevalence estimates adjusted by both the sensitivity and specificity of progressively stringent ground truths definition (Supplementary Table 3) are shown in Fig. 2c. The likelihood method, and the independent utilisation of the baseline ground

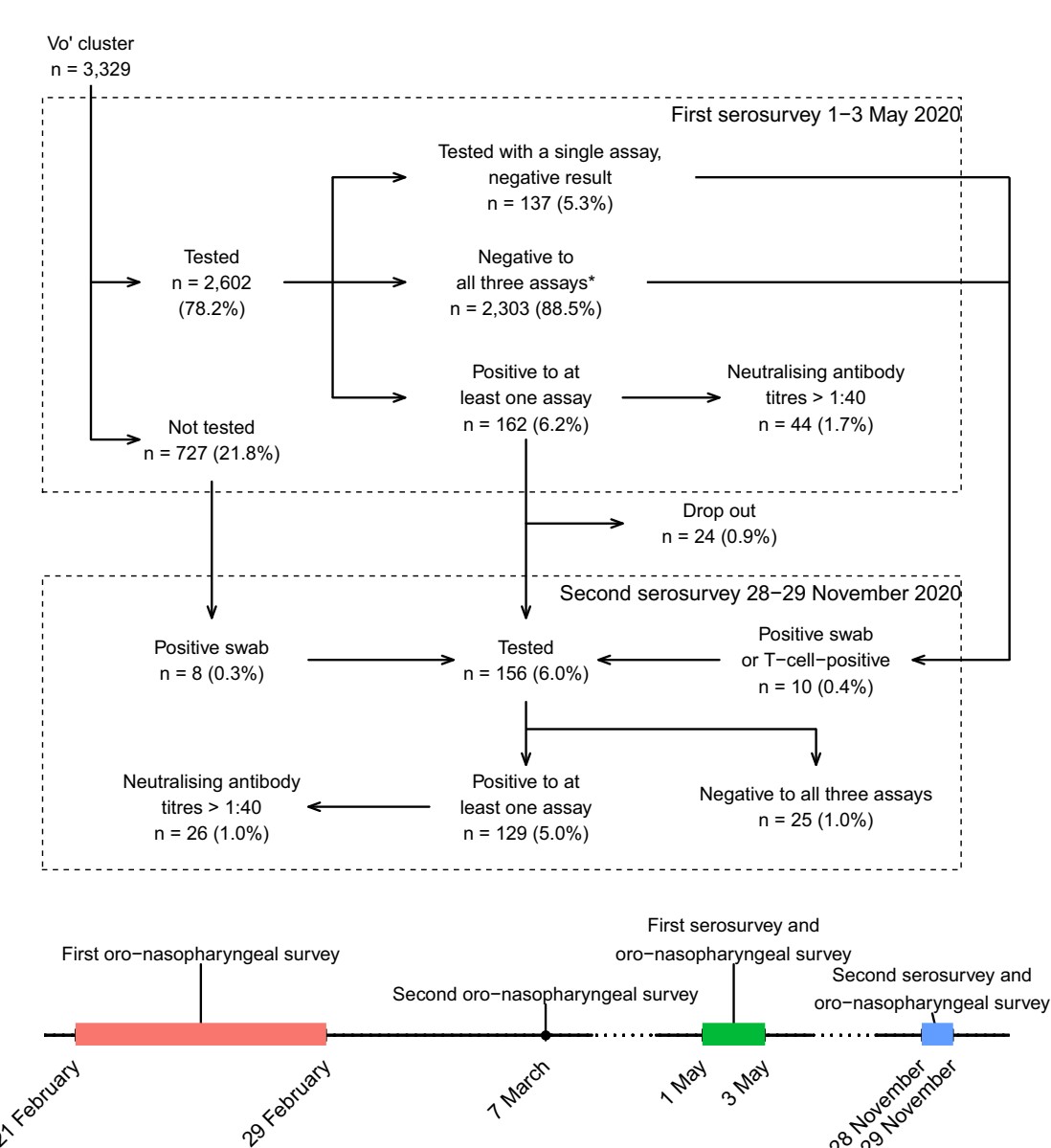

**Fig. 1 Study description. a** Flow chart illustrating the study design and the stratification according to test results rates on the occasion of the serosurveys conducted in Vo' on 1–3 May 2020 and 28–29 November 2020. **b** Timeline of the surveys conducted in the study area since the start of the SARS-CoV-2 epidemic in Vo'.

| Table 1 Commercial assays employed in the study to identify IgG anti-SARS-CoV-2. | | | | |
|---|---|---|---|---|
| **Test** | **Manufacturer** | **Recognised antigen** | **Method** | **Manufacturers' thresholds** |
| LIAISON® SARS-CoV-2 S1/S2 IgG | DiaSorin | S1/S2 | CLIA[a] | Negative: <12.0 AU/mL<br>Equivocal: $12.0 \leq x < 15.0$ AU/mL<br>Positive: ≥15.0 AU/mL |
| Elecsys® Anti-SARS-CoV-2 | Roche | N | ECLIA[b] | Positive: <1.0<br>Negative: ≥1.0 |
| ARCHITECT® SARS-CoV-2 IgG | Abbott | N | CMIA[c] | Negative: <1.4<br>Positive: ≥1.4 |

[a]Chemiluminescence immunoassay.
[b]Electro-chemiluminescence immunoassay.
[c]Chemiluminescent microparticle immunoassay.

truth definition, overcome the uncertainties associated with the use of the raw seroprevalence estimates adjusted by the assay-specific sensitivity and specificity obtained from the validation experiments only (Table 3 and Fig. 2b). The concordance among the assays is reported in Supplementary Table 4.

**Antibody reactivity and persistence**. Of the 125 subjects exposed to SARS-CoV-2 according to the baseline ground truth definition, 101 (80.8%) participated to the May serosurvey. Among them, 93.5% (86 out of 92, 95% CI 86.3–97.6%), 84.2% (85 out of 101,

95% CI 75.5–90.7%), and 100% (92 out of 92, 95% CI 96.1–100%) had a positive result for Abbott, DiaSorin and Roche, respectively, whereas 44.9% (44 out of 98, 95% CI 34.8–55.3%) had a neutralising titre greater than 1:40 (1/dil). In November, 86 subjects (68.8%) were tested again, all of them except one (98.8%) tested positive to at least one serological assay. Of the 81 subjects tested with all three assays, 25 (30.9%) tested positive to all three assays and 1 (1.2%) tested negative to all three assays. The longitudinal assessment of antibody titres over a seven-months period shows some differences depending on the assay considered (Fig. 3). The antibody titres quantified by Roche displayed a non-significant decrease ($p = 0.09$, Wilcoxon signed-rank test), whereas for the DiaSorin, Abbott, and micro-neutralisation assays, we found a statistically significant decrease between the first and second serosurvey (Fig. 3) (DiaSorin $p = 0.03$; Abbott $p < 0.0001$; micro-neutralisation $p < 0.0001$, Wilcoxon signed-rank test). Notably, in November, 16 subjects showed an antibody titre more than double that observed in May (4 detected by DiaSorin, 12 detected by Roche, and 1 by micro-neutralisation; 1 detected by both DiaSorin and Roche assays). A substantial fraction of these

| Table 2 Tested individuals and positivity rates for the different assays in the Vo' cluster in May 2020. | | |
|---|---|---|
| **Serological test** | **Tested** | **Positive (%)** |
| Abbott | 2,457 | 95 (3.9%, 95% CI 3.1–4.7%) |
| DiaSorin | 2,594 | 135 (5.2%, 95% CI 4.4–6.1%) |
| Roche | 2,457 | 98 (4.0%, 95% CI 3.3–4.9%) |
| Oro-nasopharyngeal swab | 2,599 | 0 (0%, 95% CI 0.0–0.0%) |

| Table 3 Assay-specific performance estimates. | | | | |
|---|---|---|---|---|
| **Data source** | **Parameter** | **Abbott** | **DiaSorin** | **Roche** |
| Validation experiment only | Sensitivity | 0.939 (0.797–0.993)[a] | 0.852 (0.767–0.914)[b] | 0.939 (0.797–0.993)[a] |
| Validation experiment only | Specificity | 1.00 (0.934–1.00)[c] | 1.00 (0.832–1.000)[d] | 0.976 (0.874–0.999)[e] |
| Validation experiment & serosurvey results | Sensitivity | 0.969 (0.926–0.994) | 0.856 (0.801–0.904) | 0.968 (0.924–0.994) |
| Validation experiment & serosurvey results | Specificity | 0.997 (0.994–0.999) | 0.982 (0.976–0.987) | 0.997 (0.994–0.999) |
| Validation experiment & serosurvey results | PPV | 0.906 (0.855–0.951) | 0.627 (0.539–0.720) | 0.916 (0.866–0.963) |
| Validation experiment & serosurvey results | NPV | 0.999 (0.999–1.000) | 0.995 (0.992–0.998) | 0.999 (0.999–0.999) |

Mean and 95% CI of the sensitivity and specificity obtained from the validation experiments only (VEO) and from the combined validation experiments and serosurvey results reported in Table 2. The sample sizes are given in the footnote. PPV positive predictive value; NPV negative predictive value. The 95% CrI around the PPV and NPV have been estimated by bootstrapping.
[a]33 SARS-CoV-2 PCR positive tested ≥ 30 days post symptom onset, of which 31 seropositive.
[b]101 SARS-CoV-2 PCR positive tested ≥ 30 days post symptom onset, of which 86 seropositive.
[c]54 SARS-CoV-2 PCR negative tested, of which 54 seronegative.
[d]20 SARS-CoV-2 PCR negative tested, of which 20 seronegative.
[e]42 SARS-CoV-2 PCR negative tested, of which 41 seronegative.

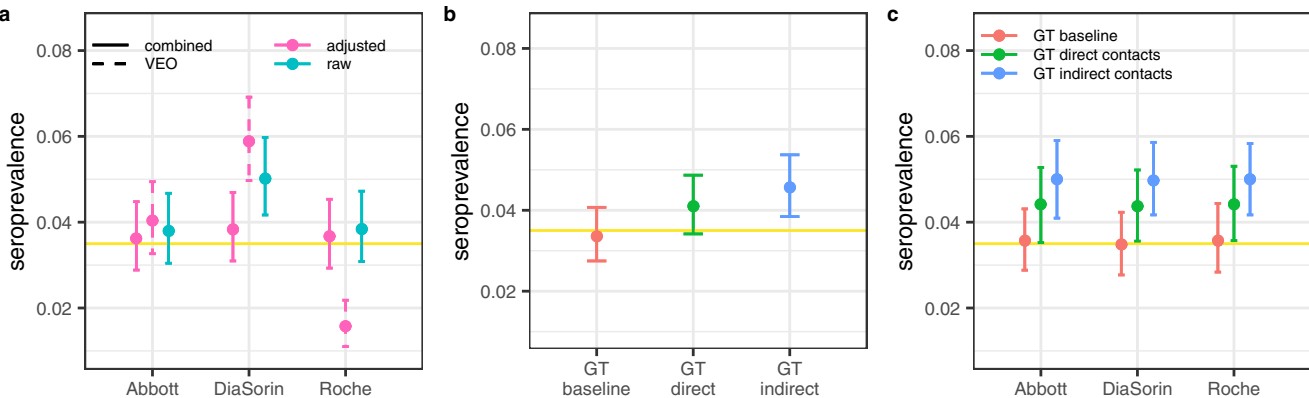

**Fig. 2 Seroprevalence estimates in Vo' in May 2020. a** Assay-specific seroprevalence estimates: the raw seroprevalence estimates (cyan) represent the proportion of subjects testing positive; adjusted seroprevalence estimates (pink) are the raw seroprevalence estimates adjusted for the estimated assay-specific sensitivity and specificity (Table 3) from the validation experiments only (VEO) (dashed) and the results of the combined analysis of validation experiments and serosurvey results (solid); **b** Estimated seroprevalence in May 2020 using the three ground truth definitions: baseline (red), direct contacts (green) and indirect contacts (blue). **c** Assay-specific seroprevalence estimates adjusted for the sensitivity and specificity of three ground truth (GT) definitions (Supplementary Table 3). In each panel, the horizontal solid yellow line represents the mean seroprevalence estimate obtained from the multinomial likelihood model; the horizontal yellow shading represents the 95% CrI. Points represent the mean, the error bar identifies the 95% CrI. The results were obtained by sampling $n = 1,000$ realisations from the posterior distribution of the parameters.

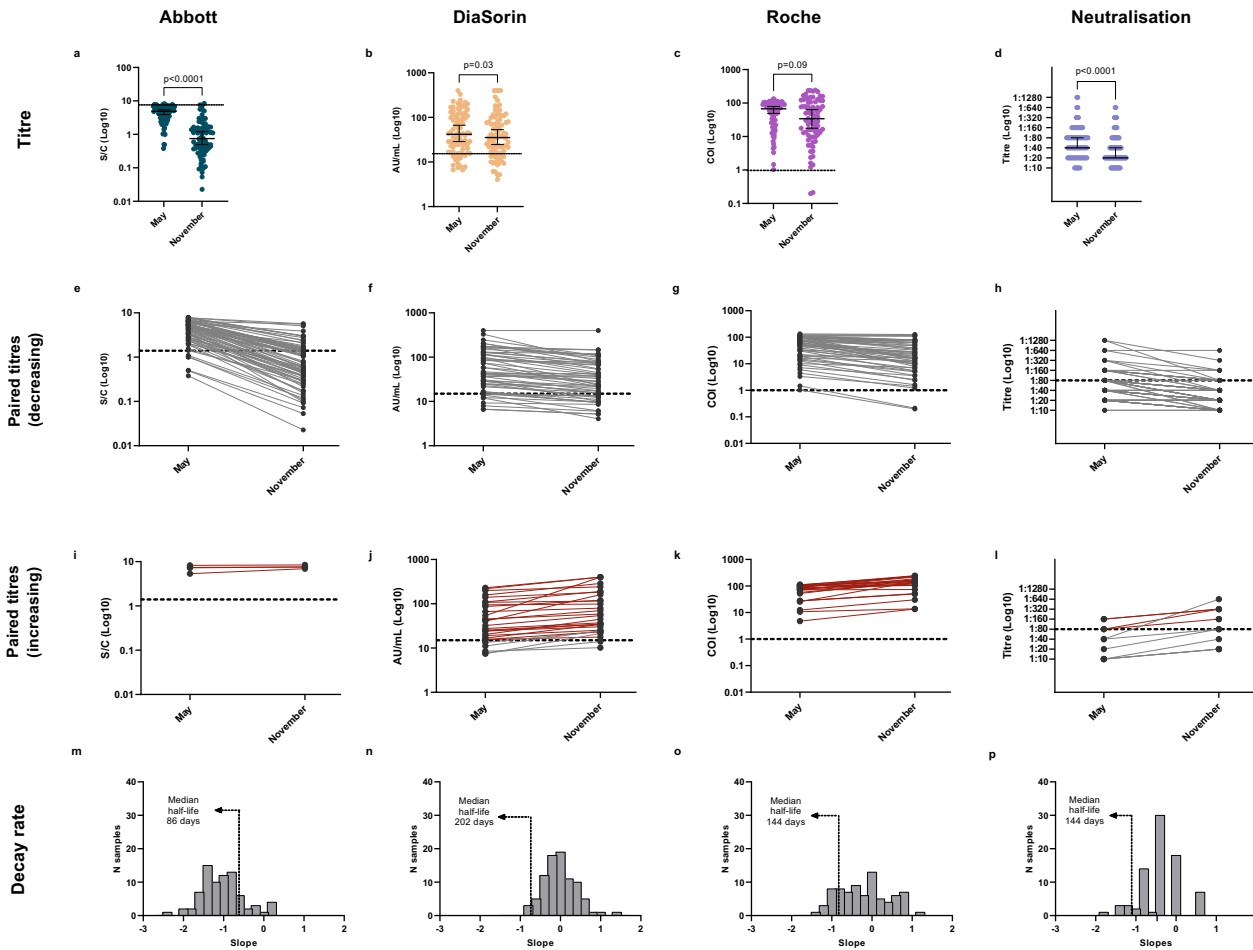

**Fig. 3 Antibody titres and dynamics at nine-month post-infection follow up. a–d** Observed antibody titres among subjects testing positive in May and re-tested in November by Abbott (*n* = 78), DiaSorin (*n* = 86), Roche (*n* = 78), and micro-neutralisation assays (84), respectively (two-sided Wilcoxon signed-rank test). Only subjects belonging to the baseline ground truth definition have been included. The horizontal line represents the median, the vertical line represents the 95% confidence intervals. **e–l** Antibody dynamics of individual subjects testing positive between May and November, by Abbott, DiaSorin, Roche, and micro-neutralisation. 88.9% (64 out of 72, 95% CI 79.3–95.1%), 20.5% (15 out of 73, 95% CI 12.0–31.6%), and 35.9% (28 out of 78, 95% CI 25.3–47.6%) of individuals showed a reduction of antibody titre of 50% or more by November when tested with Abbott, DiaSorin, and Roche, respectively, whereas 34.7% (25 out of 72, 95% CI 23.9–46.9%), 89.0% (65 out of 73, 95% CI 79.5–95.1%), and 97.4% (76 out of 78, 95% CI 91.0–99.7%) of subjects were still positive in November. For the micro-neutralisation assay, 41.0% (16 out of 39, 95% CI 25.6–57.9%) of subjects had their neutralising titres decreased by at least two dilution factors by November, and 46.1% (18 out of 39, 95% CI 30.1–62.8%) who had a titre greater than 1:40 (1/dil) in May still preserved that level by November. Subjects with an increasing trend between the two timepoints are shown separately in panels (**i–l**), where red lines highlight individuals who tested positive in May and showed an increase in their antibody titres by November (38 subjects in total, 1 for Abbott, 24 for DiaSorin, 23 for Roche, and 4 for micro-neutralisation). **m–p** Distribution of the estimated antibody decay rates. Among the subjects with a positive serological test result in May and excluding the subjects with a doubling antibody titre between May and November, we estimated a median half-life of 86 (95% CI 73–94) days, 202 (95% CI 140–270) days, 144 (95% CI 114–217) days, and 144 days (95% CI 117–178) for the antibodies detected by the Abbott, DiaSorin, Roche, and micro-neutralisation assays, respectively.

individuals (9 out of 16, i.e., 56.3% 95% CI 29.9–80.2%) reported likely re-exposure events to SARS-CoV-2. Excluding the subjects with a doubling antibody titre between May and November, the median half-life of the antibodies detected by Abbott, DiaSorin and Roche are of 86 days (95% CI 73–94), 202 days (95% CI 140–267) and 144 days (95% CI 114–214), respectively.

**Correlation between antibody detection assays**. Serological and micro-neutralisation titres obtained from sera collected in May and November from exposed individuals according to the baseline ground truth definition were analysed using Pearson's marginal and partial correlations (Supplementary Fig. 1). At both timepoints, we found strong relationships by partial correlation only between serological assays targeting the N protein (Abbott and Roche, *r* = 0.77, *p* < 0.01 and *r* = 0.82, *p* < 0.01 in May and

November, respectively), and between tests capturing the antibody response against the S antigen (DiaSorin and micro-neutralisation *r* = 0.50, *p* < 0.01 and *r* = 0.75, *p* < 0.01 in May and November, respectively).

**Association analysis**. Significant differences were observed in the mean antibody titres across age group and by BMI categories using all assays (Fig. 4; Supplementary Table 5), with a significant difference in the antibody titres quantified by the Abbott assay in May between underweight and obese BMI categories (*p* = 0.03, Tukey's Honest Significant Difference). Using linear regression, we observed a significant association between antibody titres and BMI among symptomatic infections and no significant association among asymptomatic infections (Supplementary Table 5). We found significant differences in the antibody decay rates by

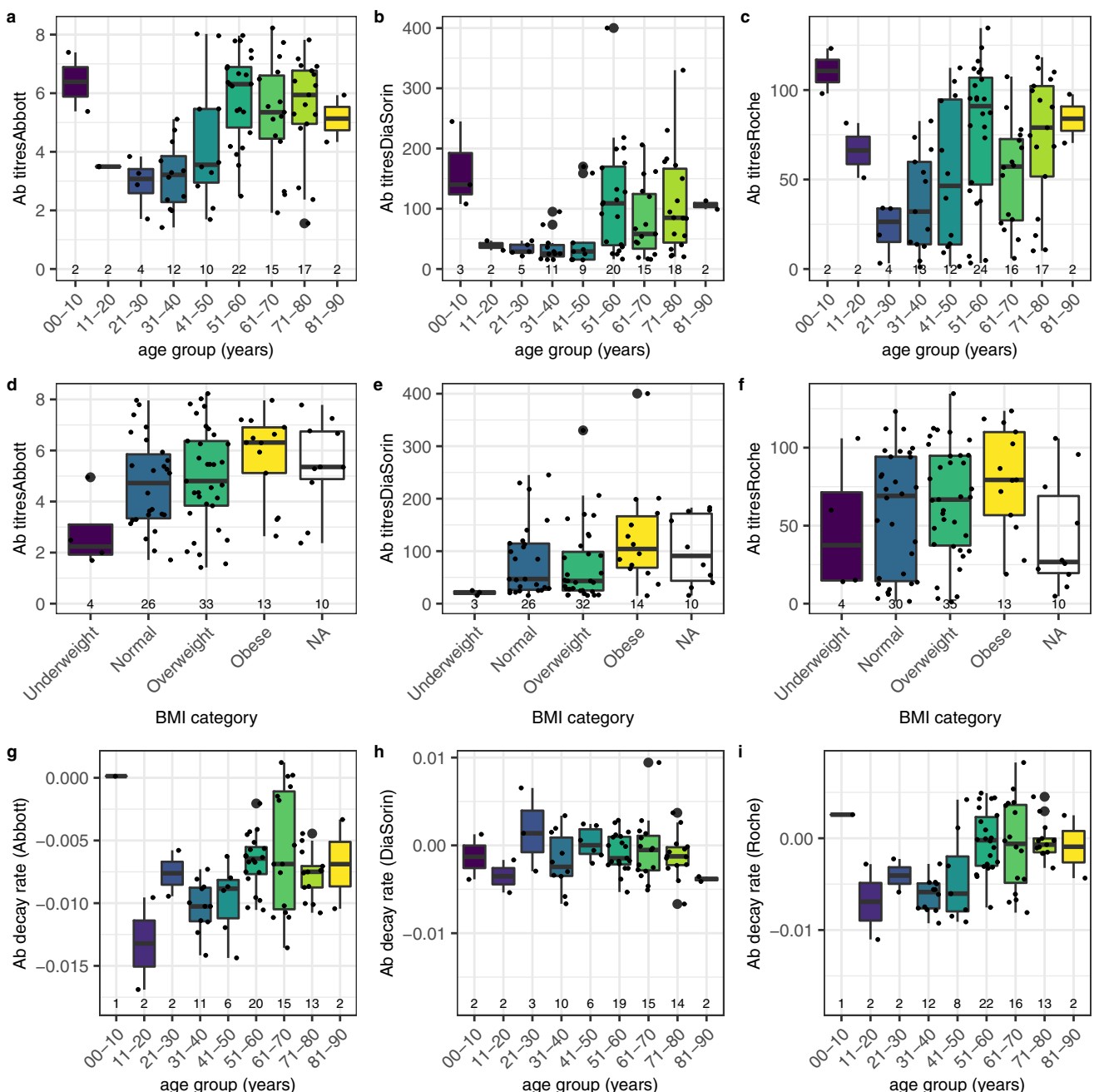

**Fig. 4 Association between antibody titres and antibody decay rate with age group and BMI category.** Antibody titre distribution by age group (**a**–**c**) and by BMI (**d**–**f**) and the estimated antibody decay rate distribution by age group (**g**–**i**) according to Abbott, DiaSorin, and Roche assays, respectively. In all panels, bold is the median, box is the interquartile range, whiskers define the range having removed the outliers; outliers are defined as observations further from 1.5 times the interquartile range. The association analysis was conducted on the subjects identified as exposed to SARS-CoV-2 according to the baseline ground truth definition. The numbers at the bottom of each panel represent the number of subjects in each category.

age using the Abbott and Roche assays (Fig. 4; Supplementary Table 6). We found no statistically significant differences in the average antibody decay rate between BMI categories (Supplementary Table 6). However, we observed a significant association between the decay rate of the antibodies quantified by Roche and the BMI among symptomatic infections, with 0.0003 decrease in antibody decay rate per unit increase in BMI (Supplementary Table 6). Additional results are discussed in Supplementary Note 1, and we report the association between symptoms occurrence and comorbidities in Supplementary Table 7 and the association between symptoms occurrence and medical treatment in Supplementary Table 8.

**Within-household SARS-CoV-2 transmission**. Using the baseline ground truth definition, we identified 1,118 households in Vo' with known infection status for all household members (Supplementary Table 9). Overall, 5.01% (95% CI 3.81–6.46%) of the households in Vo' experienced at least one SARS-CoV-2 infection. The household size distribution, along with the observed attack rates and frequency of SARS-CoV-2 infections by age group and household size are shown in Supplementary Fig. 2. Using the model proposed by Fraser et al.[24] (summarised in Supplementary Methods, section 4) we found that models allowing for overdispersion in the offspring distribution (model

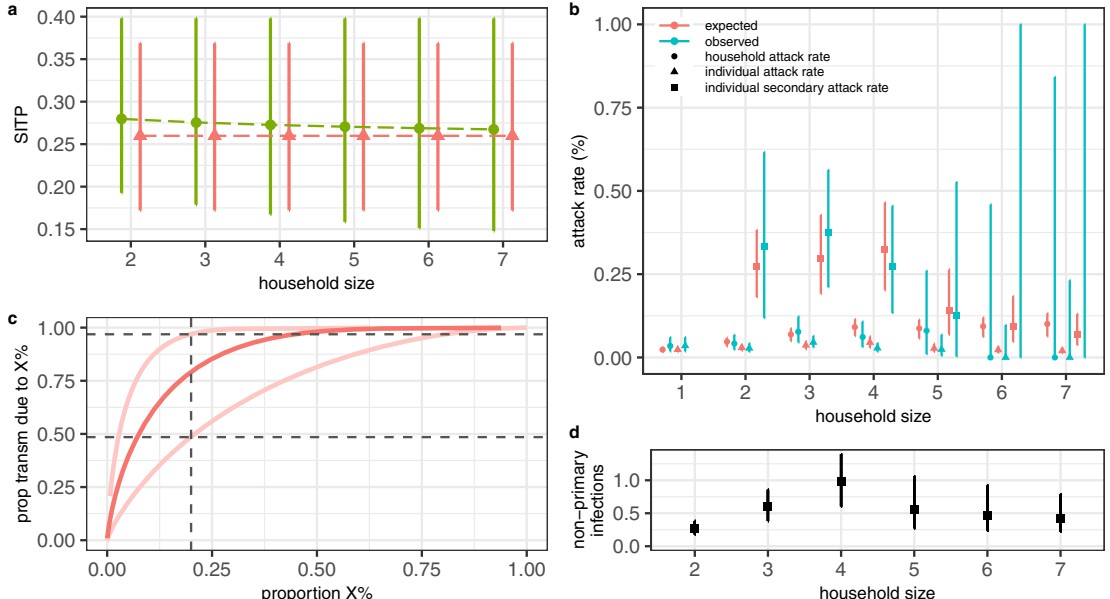

**Fig. 5 Within-household SARS-CoV-2 transmission estimates. a** Mean (points) and 95% CrI (lines) of the Susceptible-Infectious Transmission Probability (SITP) estimates by household size obtained with model V (red) and PV (green). **b** Mean (points) and 95% CI (lines) of the observed household, individual-level and secondary attack rates compared to the mean and 95% CrI of the estimated attack rates obtained with model V. **c** Median (dark) and 95% CrI (light) proportion of transmission (y-axis) attributable to the most infectious proportion of infections (x-axis) obtained with model V; homogeneity in transmission would result in y = x. **d** Mean (points) and 95% CrI (lines) of the estimated number of non-primary infections by household size obtained with model V. In all panels, the results were obtained by sampling $n = 1,000$ realisations from the posterior distribution of the model parameters.

variants V) are favoured in terms of the Deviance Information Criterion (DIC) (Supplementary Fig. 3). Neither of the model extensions explored, including an alternative interpolation between frequency and density dependent transmission by household size and the existence of high- and low-transmission groups improved the model fit (Supplementary Fig. 3). Figure 5 shows the fit of model V, which is the most parsimonious amongst the selected models (i.e., the models with the lowest DIC), to the observed household, individual and secondary attack rates. Using model V, we estimate an escape probability from sources of infection outside the household of 97.8% (95% CrI 97.0–98.2%) (or equivalently, that the probability of getting infected from someone outside the household is 2.2% (95% CrI 1.8–3.0%)), a within-household Susceptible Infectious Transmission Probability (SITP) of 26.0% (95% CrI 17.2–36.9%) and that 79.1% (95% CrI 48.9–98.9%) of SARS-CoV-2 transmission is attributable to the 20% most infectious individuals (Fig. 5). All parameter estimates are provided in Supplementary Table 10.

**Sensitivity and impact of contact tracing.** The availability of extensive contact tracing data and the parallel independent identification of infected subjects by mass swab testing allow for a quantitative assessment of the sensitivity of contact tracing and its impact on the epidemic final size. During the first wave of SARS-CoV-2 transmission, contact tracing identified and isolated 44 PCR positive named contacts, out of the 100 PCR positive subjects identified in February and March 2020 (Fig. 6a) belonging to the Vo' cluster (see Methods). To quantify and disentangle the relative impact of contact tracing from the impact of mass testing and lockdown on the observed epidemic dynamics, we extended the transmission model developed in Lavezzo et al.[22] to explicitly include contact tracing (Supplementary Fig. 4). A detailed description of the transmission model is given in the Supplementary Methods, section 5. We fitted the model to the observed prevalence of infection among traced contacts (44 out of 190) and

sensitivity of contact tracing (44 out of 100) (Fig. 6b, c) as well as to the observed prevalence of infection in the study population stratified into asymptomatic, pre-symptomatic and symptomatic subjects (Fig. 6d). Summary statistics of the posterior distributions of the parameters are given in Supplementary Table 11. Our results show that, assuming an initial $R_0$ of 2.4, mass testing and lockdown had a major effect on epidemic control, reducing the basic reproduction number by an average of 83% (95% CrI 65–100%) (Supplementary Table 11). This finding is confirmed by the small relative reduction in the epidemic final size compared to the unmitigated epidemic obtained when simulating contact tracing in the absence of mass testing and lockdown (Fig. 6f). On the other hand, the implementation of contact tracing jointly with mass testing and lockdown enhanced epidemic suppression and additional contact tracing efforts would have further reduced the estimated epidemic final size (Fig. 6g). In the absence of mass testing and lockdown, our results suggest that epidemic suppression comparable in size to that observed during the first wave would have been obtained with a rate of contact tracing and isolation at least four times that implemented during the first wave (Fig. 6h).

## Discussion

Comparing population-level seroprevalence estimates across settings and locations is challenging due to potential differences in the sampled population, assay used, laboratory methods and statistical analyses performed. This study provided the unique opportunity to test differences in seroprevalence estimates solely due to the assay used. Combining multiple test results and assay validation experiments through modelling provide more accurate estimates of exposure than by inferring the seroprevalence on individual assays and using validation experiments only. The resulting population-level seroprevalence estimate of 3.5% (95% CrI 2.8–4.3%) (Fig. 2) is in good agreement with the baseline ground truth definition. We find that the probability of infection

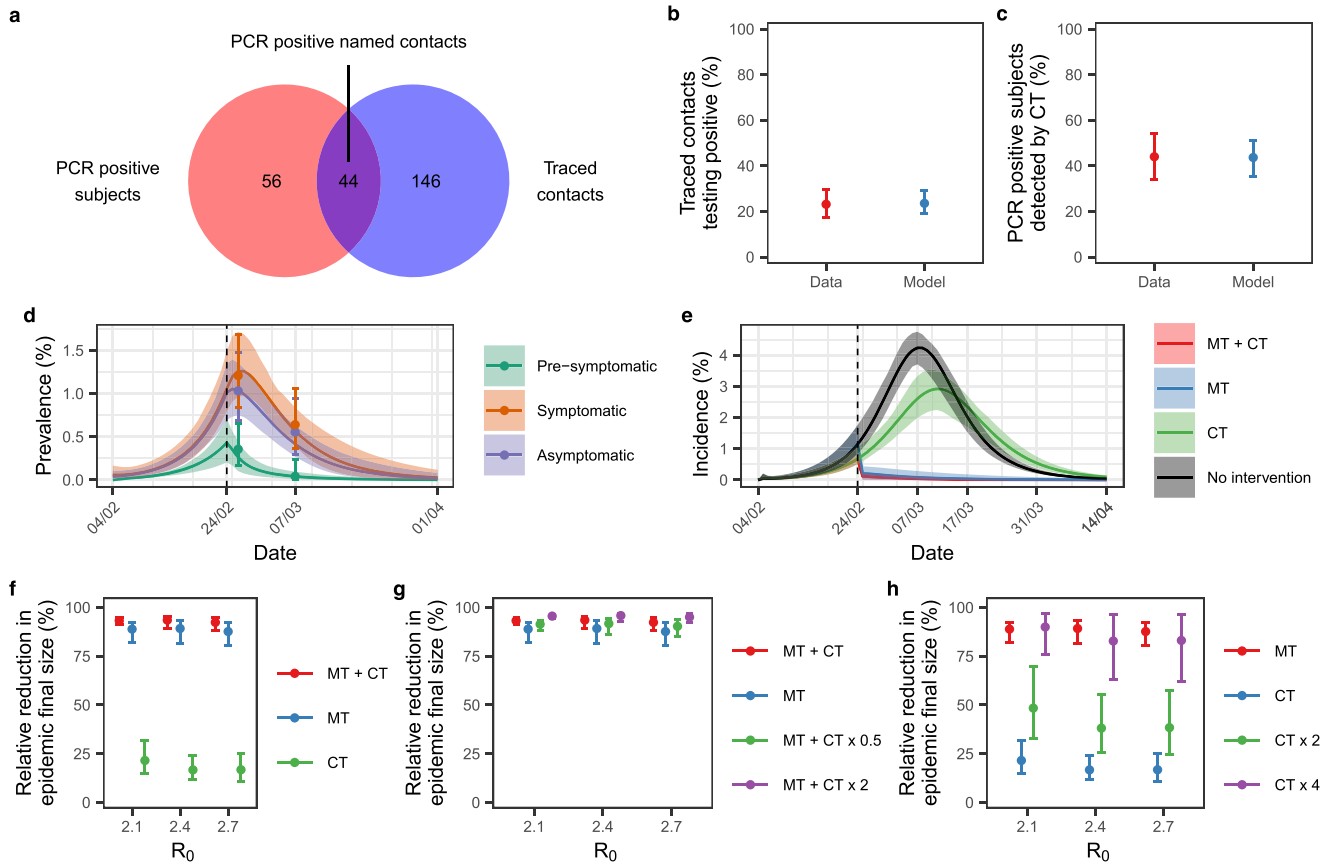

**Fig. 6 Impact of contract tracing on the epidemic dynamics. a** PCR-positive subjects identified by mass testing (red) in comparison to all subjects contacted by contact tracing (blue) and the positive traced individuals (intersection). **b** Observed (red, mean 95% exact binomial CI) and estimated (blue, mean and 95% CrI) proportion of traced contacts testing positive by PCR (**c**) and proportion of PCR positive subjects detected by contact tracing. **d** Observed and estimated SARS-CoV-2 prevalence. The points represent the observed prevalence data, with the 95% exact binomial CI. The solid lines represent the mean, and the shading represents the 95% credible interval. **e** Simulated incidence of SARS-CoV-2 infection assuming the following interventions: mass testing, lockdown, and contact tracing (MT + CT, red); mass testing and lockdown (MT, blue); contact tracing (CT, green). The unmitigated (no intervention) epidemic is shown in black. The dark lines represent the mean, shading represents the 95% credible interval. In **d** and **e**, the vertical dashed line represents the time interventions were implemented. **f** Relative reduction in the epidemic final size compared to the unmitigated epidemic obtained assuming the same interventions of **e**. **g** Relative reduction in the epidemic final size compared to the unmitigated epidemic obtained assuming the implementation of mass testing, lockdown and contact tracing (MT + CT, red), mass testing and lockdown (MT, blue), mass testing, lockdown and half the estimated rate of contact tracing (MT + CT x 0.5) and mass testing, lockdown and double the estimated rate of contact tracing (MT + CT x 2). **g** Relative reduction in the epidemic final size compared to the unmitigated epidemic obtained assuming the implementation of mass testing and lockdown (MT), contact tracing (CT), contact tracing with double the estimated tracing rates (CT x 2) and contact tracing with four times the estimated tracing rates (CT x 4). In **f**, **g**, and **h** points represent the mean and bars the 95% CrI. All estimates were obtained using 100 samples from the posterior distribution of the parameters. **b**–**e** show the results obtained with $R_0 = 2.4$.

given a positive test result varies by assay, with DiaSorin showing the lowest mean positive predictive value of 62.6% (95% CI 53.9%–72.0%) (Table 3).

Our analysis confirms substantial differences in antibody persistence by assay, with the Abbott assay showing a marked decline both in antibody titres and seropositivity over a seven-months period (Fig. 3). This finding implies that several serosurveys conducted globally with the Abbott assay, including the national study conducted in Italy[25], may underestimate the true cumulative number of SARS-CoV-2 infections. Conversely, the antibodies detected by DiaSorin and Roche appear to remain at high levels for at least nine months. The observed difference in antibody decay between the Abbott and Roche assays, which target the same (N) antigen, is in agreement with previous findings and could be due to partial differences in the employed antigens and to the fact that the range of N epitopes recognised by sera might change with time[26]. Our results, which are presented without accounting for multiple testing to facilitate direct comparisons

with other publications where only comparison-wise error rate (CER) has been controlled for, show significant differences in the magnitude of the antibody response by age group and BMI category using all assays, but not by the presence or absence of symptoms. While other studies reported significantly lower IgG levels in asymptomatic versus symptomatic infections during the acute and early convalescent phase[16,27] and at the time of virus clearance[28], our findings suggest that, over the longer term, early differences in the magnitude of antibody response level off. Over a seven-months period (May – November 2020) we observed significant differences in the antibody decay rates by age group but no significant differences by symptom occurrence nor by hospitalisation status, sex or BMI. Notably, in a fraction of individuals, we observed a marked increase in the antibody levels, in a few cases paralleled by an increase in neutralisation titre. A large proportion of these individuals (56.3%, 95% CI 29.9–80.2%) reported exposure to PCR confirmed SARS-CoV-2 infections. Accordingly, while immunity induced by natural infection in

convalescent and recovered individuals protects from the development of symptomatic disease, it can allow sufficient viral replication to boost antibody production. This has important implication for understanding the protection elicited by natural infection as well as vaccines and could potentially imply that while immunisation (either via natural infection or vaccination) protects against disease, it may allow for some level of viral replication and thus onward transmission.

The approach used to estimate the probability of within-household SARS-CoV-2 transmission provides a useful framework to understand how demographical and epidemiological factors influence transmission within the household setting. In agreement with previous reports[29–31], we estimate considerable heterogeneities in the number of secondary infections generated by an infection (k parameter, Supplementary Table 10), which poses challenges for disease surveillance and control. To the best of our knowledge, we provide the first estimate of the SITP for SARS-CoV-2, which represents a more nuanced measure of within-household transmission intensity than the within-household secondary attack rate which, by definition, does not account nor control for multiple introductions from outside the household and can thus potentially overestimate the actual extent of transmission occurring within the household setting. Among the strengths of this study is the availability of extensive contact tracing and mass swab testing data conducted independently and in parallel at the beginning of the epidemic, which allows for a detailed analysis of the contact tracing sensitivity, efficiency and impact on the epidemic dynamics and final size. In our counterfactual analysis we show that contact tracing alone, in the absence of mass testing, would have had a limited impact on the final size of the epidemic. This finding is particularly important in the light of the efforts and resources invested so far in contact tracing to limit transmission and the additional challenges posed to contact tracing in highly populated settings. Our modelling study suggests that, in the absence of mass testing, substantial increases in contact tracing efforts could suppress the epidemic to a similar extent to what has been obtained in Vo' with mass testing. From a practical perspective however, the substantial contact tracing efforts deployed in Vo' at the start of the pandemic suggest that the efficiency of traditional contact tracing, based on people's recollection and reporting, likely reached its maximum in this context. The recent detection of new, more transmissible[32] and more severe[33] SARS-CoV-2 variants in the UK and elsewhere, reinforces the urgency of improving control strategies[29,34] including widespread testing and digital contact tracing, to keep SARS-CoV-2 incidence at low levels globally.

## Methods

**Ethical approval statement.** The first and second serosurveys of the Vo' population were approved by the Ethics Committee for Clinical Research of the province of Padova (May survey approved on 30th April 2020, protocol number 0026971; November survey approved on 11th November 2020, protocol number 0068830). Study participation was by consent. For participants under 18 years of age, consent was provided by a parent or legal guardian.

### Laboratory methods

*Oro-nasopharyngeal swabs.* Upper respiratory tract samples were collected by first inserting the swab into the posterior pharynx, rubbing over both tonsillar pillars and posterior oropharynx, and then into both nostrils for about 2 cm, gently rotating against the nasal wall. The presence of SARS-CoV-2 genomes was evaluated with an in-house real-time RT–PCR method targeting the envelope gene (E), according to Corman et al.[35] (Supplementary Table 12).

*Serum samples.* Blood samples were collected in 5 ml BD Vacutainer Serum Separation Tubes (SST), centrifuged for 10 min at 1000–1300 RCF (g) and immediately frozen at −80 °C, until the execution of the serology tests to evaluate the presence of IgG anti-SARS-CoV-2. These were performed over the following weeks using three commercial kits produced by Abbott[36], DiaSorin[37], Roche[38] (Table 1) by trained laboratory staff. We quantified one antibody titre per serum sample, using the thresholds provided by the manufacturers (Table 1). We validated the assays' performance provided by the manufacturers using in-house experiments[23]. Although the Abbott and Roche tests are not quantitative according to the manufacturers' specifications, we previously demonstrated the repeatability of both methods over different days and using different batches[23].

*Micro-neutralisation assay.* Serum samples were heat-inactivated for 30 min at 56 °C, then diluted 1:10 with DMEM FBS Free medium and filtered (0.22 μm pore size). 96-wells microplates were prepared by mixing in each well 50 μl of a third passage viral isolate (GenBank accession MW468415), diluted in DMEM FBS Free to the final concentration of 100 median tissue culture infective dose (TCID50), with an equal volume of two-fold serial dilutions of sera. The serum-virus mixture was incubated for 1 h at 37 °C in a humidified atmosphere with 5% $CO_2$. After incubation, 100 μL of VERO E6 cell suspension, previously detached in DMEM 6% FBS, were added to each well and further incubated at 37 °C. Cytopathic effect was evaluated after 72 h. Then, the supernatant was removed and 120 μl of 5% formaldehyde Gram's crystal violet 40% m/v were added to each well. After 30 min of incubation, the microplates were washed with water, allowed to dry, and the absorbance was read at 595 nm. The highest serum dilution showing an optical density (OD) value equal or greater than 90% of the control sera was considered as the neutralisation titre.

### Statistical methods and mathematical modelling

*Antibody decay rate and half-life estimates.* We estimate the individual-level antibody decay rate as the change in antibody titres observed between May and November (within the same subject) divided by the number of days between the two serosurveys (212 days). The antibody half-life was estimated as the natural logarithm of 0.5 divided by the antibody decay rate.

*Seroprevalence estimates.* Denote the sensitivity and specificity as *se* and *sp*, respectively. Denote the raw seroprevalence as $serop_{raw}$, defined as the proportion of samples testing positive (number of positive/number tested). We calculate the adjusted seroprevalence $serop_{adj}$ using formula

$$serop_{adj} = \frac{serop_{raw} + sp - 1}{se + sp - 1} \qquad (1)$$

where $se + sp > 1$.

*Likelihood-based seroprevalence estimate.* We adjust the observed raw seroprevalence with respect to the assay-specific sensitivity and specificity estimated from in-house validation experiments only and using the assay-specific sensitivity and specificity estimated from the model combining the validation experiment and serosurvey results. To infer the true prevalence of infection, we assumed that data from the in-house validation experiments came from a binomial distribution and that the observed results (in terms of case counts in each of the possible assay results categories, Table 2) came from a multinomial distribution, having assumed assay independence. Full details on the likelihood formulation and methods used for statistical inference are provided in the Supplementary Methods, section 3.

*Ground truth-based seroprevalence estimate.* In a separate analysis, we adjust the observed raw seroprevalence with respect to the ground truth definition-specific sensitivity and specificity (Supplementary Table 3).

*Association analysis.* We assessed univariate normality using the Shapiro test and within group equality of variance using the Levene test. Differences in mean antibody titres and antibody decay rates between two categories (e.g., symptomatic versus asymptomatic infections, hospitalised versus non-hospitalised infections, females versus males) were assessed using the t-test when normality was supported, otherwise using the non-parametric Wilcoxon signed-rank test. Similarly, we assessed differences across multiple categories (e.g., age groups and BMI categories) using the one-way Anova test when normality was supported, otherwise using the non-parametric Kruskal-Wallis test. In the instances where less than 3 observations per category were reported, we tested the association having included and having excluded these categories (those including less than 3 observations), separately. Association was assessed using univariate and multiple linear regression. The association between symptom occurrence and comorbidities/medical treatment was assessed using Fisher's exact test. We assumed a significance level of 0.05 throughout.

*Impact of contact tracing.* We couple the official contact tracing records compiled by the local health authority with the data collected in this and previous[22] surveys to calculate the sensitivity of contact tracing $se_{ct}$, defined as the proportion of named contacts contacted and isolated by contact tracing relative to the total number of PCR positive subjects detected in February and March 2020.

We extended the transmission model developed in Lavezzo et al.[22] (see Supplementary Methods, section 5 for a summary) to include contact tracing. The flow diagram of the model is given in Supplementary Fig. 4. Contact tracing was modelled by assuming that susceptible and infected subjects (at any stage during infection) could be detected and isolated at rate $ct_S$ and $ct_I$, respectively. We

allowed for two separate detection rates to reflect the simultaneous occurrence of contact tracing with mass testing, implying a higher detection probability for infected subjects compared to susceptible subjects, as suggested by the data. We assumed that traced subjects were fully isolated, i.e., isolated infected subjects did not transmit the disease onwards and quarantined susceptible subjects were protected against the infection. We assumed that all interventions started on 24th February 2020. The effect of mass testing and the lockdown was modelled through a step change in the reproduction number, with $1 - w$ representing the percent reduction in transmission intensity. We simultaneously fitted the prevalence of infection observed in the February and March surveys, the proportion of traced contacts testing PCR positive and the proportion of PCR positive subjects detected by contact tracing by the end of the epidemic. We fitted the model in a Bayesian framework, using the Metropolis-Hastings Markov Chain Monte Carlo (MCMC) method with uniform prior distributions, assuming an initial reproduction number of either 2.1, 2.4, or 2.7 and an average duration of positivity beyond the duration of the infectious period $(1/\sigma)$ equal to 4 days. All details on the transmission model are given in Supplementary Methods, section 5. In a counterfactual analysis we explored the impact of different interventions, implemented in isolation and together, on the epidemic final size. The baseline scenario, which was fitted to data, included mass testing, lockdown and contact tracing (MT + CT). We explored the following counterfactual intervention scenarios: (i) mass testing and lockdown without contact tracing (MT); (ii) contact tracing without mass testing and lockdown (CT); (iii) mass testing, lockdown and enhanced or reduced contact tracing (MT + CT, with the specified contact tracing intensity multiplier); (iv) enhanced contact tracing without mass testing and lockdown (CT with the specified contact tracing intensity multiplier). Details on the parameterisation of the counterfactual scenarios are provided in the Supplementary Methods, section 5.

**Reporting summary**. Further information on research design is available in the Nature Research Reporting Summary linked to this article.

## Data availability
The data generated in this study are available at https://github.com/ncov-ic/Vo_serology[39].

## Code availability
The code is available at https://github.com/ncov-ic/Vo_serology[39].

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

## Acknowledgements
We acknowledge Abbott, DiaSorin and Roche Diagnostics for kindly supplying reagents, with no influence on the study design and data analysis. This work was supported by the

Veneto Region. We acknowledge the Abdul Latif Jameel Institute for Disease and Emergency Analytics, funded by the Community Jameel, and the MRC Centre for Global Infectious Disease Analysis (reference MR/R015600/1), jointly funded by the UK Medical Research Council (MRC) and the UK Department for International Development (DFID), under the MRC/DFID Concordat agreement and is also part of the EDCTP2 programme supported by the European Union. ID acknowledges research funding from a Sir Henry Dale Fellowship funded by the Royal Society and Wellcome Trust [grant 213494/Z/18/Z]. E. L. and S.T. acknowledge research funding from the European Union's Horizon 2020 research and innovation programme, under grant agreement No 874735 (VEO). E.L. acknowledges funding from the University of Padova and the Department of Molecular Medicine (STARS-CoG ISS-MYTH and PRID/SID 2020). C.C. acknowledges funding from the Wellcome Trust (grant 203851/Z/16/Z). CAD acknowledges the UK NIHR for funding of the NIHR Health Protection Research Unit in Emerging and Zoonotic Infections.

## Author contributions

Study conceptualisation: I.D., E.L., A.C. Coordination of data collection and curation: E.L. Performed laboratory testing: M.P., C.B., M.C., F.S., E.F., C.D.V., M.C.V., T.Z., R.C., V.L., L.R., I.C., M.P., A.P. Sampling logistics and collection: E.L., F.C., G.C., M.N., E.N., E.S., B.L., L.F., S.G., M.F., L.M., M.G., S.M., G.A. Performed swab and blood sampling: F.C., G.C., M.N., E.N., E.S., B.L., L.F., S.G., M.F. Statistical analysis: I.D., E.L., L.M., M.G., S.T., A.R.B., C.A.D. Mathematical modelling: I.D., C.C., H.J.T.U., N.M.F, C.A.D., A.C. Funding acquisition: I.D., E.L., C.C., N.M.F., C.A.D., A.C. Methodology: I.D., E.L., A.R.B., N.M.F., C.A.D. Software: I.D., C.C. Visualisation: I.D., C.C., E.L., L.M. Writing - original draft: I.D. Writing - review & editing: I.D., E.L., C.C., H.J.T.U., A.R.B., C.A.D., A.C. Verified the underlying data: I.D., E.L., A.C.

## Competing interests

The authors declare no competing interests.
