## [Peer Review File · Nature Communications]

REVIEWER COMMENTS

Reviewer #1 (Remarks to the Author):

Thank you for the opportunity to review "SARS-CoV-2 antibody dynamics, within-household transmission and the impact of contact tracing from community-wide serological testing in the Italian municipality of Vo." The authors should be commended for such deep analyses with these unique data from Vo, Italy. The authors have combined three fairly distinct analyses into one paper: (1) analyses related to the persistence and concordance of different immunoassays, (2) household modeling analyses to estimate the susceptible infectious transmission probability and the cumulative inter-household transmission risk and (3) an modeling exercise to explore the impact of contact tracing of the 100 PCR-confirmed cases in Vo. All three of these analyses are interesting and relevant but combining them all together made this paper really challenging to read (for this humble reviewer at least). The methods, supplement and results do a lot of cross referencing that I was only able to really grasp what the authors were doing when using 2-3 viewing panes at the same time.

Antibody Persistence and Seroprevalence:

- In Figure 2, it appears that the GT baseline assumption results align best with the yellow 'mean seroprev estimate.' Is this yellow estimate using the GT baseline assumption as part of the sensitivity estimation? If so, seems a bit circular? If not, would be helpful to clarify exactly what goes into the yellow.
- The Abbott test is not a quantitative test yet the authors seem to use the readouts as precise numeric values that are comparable between batches. I could see using a standard curve to translate this into relative antibody units but I do not think the authors did this. I suggest motivating this strong assumption or limiting analyses based on numeric readouts to only the quantitative tests. It should also be noted that the Roche N assay has a truncated upper end of the dynamic range which you may want to consider (e.g., see figure in this recent pre-print <https://www.medrxiv.org/content/10.1101/2021.03.16.21253710v1>).
- The authors do a lot of statistical tests on this fairly small dataset to look for differences by age, sex, BMI etc in antibody levels and decay. I suspect that accounting for multiple testing will lead to a fair number of these 'significant' differences going away. Also, I think this section of association analyses could be shortened/simplified. The power to detect differences in most of these analyses is super low and calling out each time a difference wasn't detected might not be necessary if its in a table/plot somewhere.
- Lines 146-148 - I don't see any methods for how half-life was estimated.
- It's not totally clear to me what data from this study are being used within the Lloyd-Smith model. It would be helpful to clarify how this is connected to the data in this study.

Household Transmission:

- Great to see an estimate of SITP here and all the model variants considered!
- From the table in the appendix it seems like model V is not the one with the lowest DIC as suggested in the main text (line 192) nor are the model variants V the ones with the lowest DICs (ln 187). Seems like PVXZ and PVZ have the lowest though the Δ DIC is quite small. This might be a good case for model averaging or simply going with the one with the lowest DIC.
- Since most users won't know what SITP is, I would give a more interpretable definition when providing the result on line 194.
- Is there any reason you don't mention the Q parameter in the results?
- The

Contact Tracing Analysis:

- It isn't clear to me exactly what was done here and whether the conclusions that contact tracing probably made little impact are justified. Perhaps if I read the referenced Lavezzo et al paper, this would clarify but this should not be expected of the average reader. My gut is that this analysis should be removed from this paper but if not, I would do a lot more to explain the model, how the data were actually used to fit the model (may be simply through se_{ct} ?). With clustering in who is traced and clustering in infections (as suggested by the 80/20 stuff) I would have expected to see at least a larger range of results (in addition to a larger modal effect).

Editorial/Minor Comments:

- Ln 39 - I assume this is the SITP. The way in which it is described is not really accurate in the abstract. I suggest rephrasing to be more precise about what this 27.3% means.
- Line 59 - should be "from an *infectious* household member"
- Lines 69-71 - please cite references for the 'disappointing results'
- Line 113 of supplement: I suggest using cumulative incidence in place of prevalence for θ .
- Figure 4 - should the final sentence of the caption refer to panels a and b rather than b and c?

Reviewer #2 (Remarks to the Author):

Who am I?

My name is Alex Washburne and non-anonymous reviews can add the 'peer' in peer review for more respectful discussion and assist the authors in understanding the limitations of their reviewer (there are many!). While I have background ranging from immunology to mathematics, ecology and epidemiology, I know enough about immunology to know that I'm not properly equipped to provide comprehensive reviews on the comparisons of different serological assays, different antigens of different SARS-CoV-2 proteins, and more. Consequently, I'll focus my review efforts on where I may have the most insight: the statistics, modelling, and placement of this article in the broader contexts to which it's connected.

Overview

The authors present empirical results from extensive nasopharyngeal PCR and serosurvey surveillance of the small and relatively closed population of Vo, Italy. The authors also monitored close-contacts of confirmed cases.

The authors estimate the decay (or, in some cases, amplification) of antibody titers after initial confirmed infections and explain variation in waning immunity based on age, BMI, and symptoms of patients. The authors also explain variation in secondary attack rate with household size. Finally, the authors combine these empirical results into estimates of the impact of contact tracing on epidemic suppression.

Overall, I feel this paper is well-written and a valuable addition to the literature. My main comment is the need to reconcile the oropharyngeal/nasopharyngeal swab discrepancies between this and Lavazzo et al.'s previous description of the Feb/March survey (and possibly contending with variable false-negative rates). As it is, these two papers tell different methods for sample collection that make me question which method was in fact used to swab patients in Vo in Feb/March 2020 (Lavazzo et al. describe oropharyngeal swabs of both tonsillar pillars and the oropharynx, whereas this study describes collection of nasopharyngeal swabs; others have reported high false-negative rates of oropharyngeal swabs that aren't addressed when interfacing the model with the swab data). Otherwise, I provide comments on some presentations of models & figures that I feel can reduce friction & increase information content of this paper, but these aren't as necessary as the swab issue above.

Major Comments

Use of cumulative incidence vs. seroprevalence

Relationship between percent of population infected and seroprevalence: The y-axis of figure 2, and text relating to figure 2, discusses the "percent of the population infected" as interchangeable with seroprevalence (or seroprevalence adjusting for false-negative rates of serological assays?). However, surveys in the UK nurses (using Oxford ImmunoTec's assays) have revealed a large body of patients who are infected but don't seroconvert and are identified by T-cells recognizing SARS-CoV-2 specific antigens. I would encourage integrating language about this limitation throughout, referring to "seroprevalence" instead of "percent of population infected" unless explicit adjustments based on T-cell surveys and failure of mild/asymptomatic patients to seroconvert are conducted. Most Se/Sp specs from ground-truthing of serological assays come from studies of confirmed cases, and this may be biased towards patients with severe outcomes more likely to seroconvert. The authors make what I feel is a useful contribution to this discussion by their

association analysis finding no relation between symptom severity and antibody titers (this somewhat rebuts my claims above and is in disagreement with literature - e.g. Jiang et al. "Antibody seroconversion" - showing symptom severity having a significant effect on rates of even IgG seroconversion), but that's tucked away in a supplemental table (see my comments below on figures S2 & figure 3 - I feel like some of these supplemental results may add impact to the manuscript and the authors' foundational studies of Vo if presented in the main MS).

Swabs:

If I understand this correctly, the authors treat PCR positive results today as similar to PCR positive results in February/March 2020, despite PCR tests in this paper being reported as nasopharyngeal swabs and PCR tests in the Lavazzo et al. paper using the February/March 2020 data being reported as oropharyngeal swabs. Nasopharyngeal and oropharyngeal swabs have different sensitivities & specificities (see Wang et al. below, though many others have evaluated this and estimates vary with many other sources of variation including whether the nasopharynx is swabbed through just one or both nostrils) and require either (a) amending Lavazzo et al if "oropharyngeal" swabs weren't actually used (though detailed description of swabs of tonsillar pillars and the oropharynx were included there), (b) amending this paper to clarify that oropharyngeal swabs are used throughout, or (c) considering the sensitivity of attack rate and other estimates to different swab procedures. (the authors report in this paper as collecting nasopharyngeal swabs and cite Lavazzo et al "as reported previously", but that paper reports collecting oropharyngeal swabs). Lavazzo et al. specifically describe swabbing near the tonsils; given the complicated tissue tropism and variable sensitivity of PCR tests of different tissue types, this is a necessary revision and these issues of swabs, false-negative rates, and more come up again in the model parameterization & evaluation (below).

Model:

The corroboration of seroprevalence estimates with the model in Lavazzo et al. (line 219) raised a few questions for me. First, I noticed that Lavazzo et al. did not include age structure despite an estimated 3-fold increase of PCR positivity in elderly patients reported in that paper - this evidence of major heterogeneity in transmission in Vo raises questions about a homogeneous models' accuracy in estimating epidemic size, something the authors acknowledge in trying to explain the difference between seroprevalence and model-estimated epidemic sizes but that raises the question of why they didn't subsequently adjust their estimates of epidemic size to account for this 30% increase in epidemic size in their model vs. seroprevalence estimates (3.5% seroprevalence to 4.9% model-estimated epidemic size). Second, Lavazzo et al. appear to have fit their model by assuming a zero false-negative rate for the oropharyngeal swabs used - this is not consistent with other studies documenting a sometimes very large (as high as 73%) and provider-dependent false-negative rates (see Wang et al. below; there are many studies investigating false-negative rates of swabs of different types, all of them finding a non-negligible false-negative rate). Third, the authors claim a "good" agreement between Lavazzo et al. estimates and their current seroprevalence, but a midpoint estimate of 4.9% prevalence for Lavazzo et al compared to a 3.5% seroprevalence estimate is not particularly good - the 4.9% midpoint estimate from Lavazzo et al. is outside the 95% credible interval for the serosurvey, so these are different, and that difference raises questions about deeper model-structural uncertainties in the output epidemic sizes from counterfactual simulations (if you change the model structure to account for documented heterogeneity in attack rates by age and that changes epidemic sizes by 30%, then which model structure's output should we rely on?). Fourth, the authors use an R0 estimate of 2.4, but there is considerable regional variation of R0 and others estimate higher R0 (e.g. Sanche et al. estimate an R0 as high as 5.5, which appears consistent with the 2-day doubling times in many densely populated metro areas in the US under generation time distributions reported in the literature) - was there an effort to estimate R0 for Vo? Or, should we evaluate how Vo's intensive contact tracing would play out across regions with variable R0?

Ultimately, all models are wrong, some are useful, and it seems to me the value of the model for this paper is as a means to estimate final epidemic size under various NPIs. For that end, the model is sufficiently useful and I feel the questions raised above can be avoided by tightening up the treatment of different swab types and not resting as heavily on the model's predictions as such claims raise these more sharp-penciled questions about model quality & prediction accuracy that I feel detract from a sufficiently interesting and impactful empirical paper. One could also consider how the model's predicted relative epidemic sizes under various NPIs (relative to the unmitigated

scenario) differs from simpler $1-1/((1-a)*R_0)$ calculations where a is the proportion of contacts traced & isolated.

I raise these issues primarily to provide the authors an n=1 focus group of how the model is received. To alleviate these points of friction, I recommend the following:

- 1) clarify swab collection and, if different, account for the different false negative rates of different swab types over time. This is the only change I feel is truly necessary for reproducibility.
- 2) It was the qualifier that the model was a "good" fit to seroprevalence that sent me down this hole of evaluating the model's assumptions & raising sharp-penciled questions about prediction accuracy... I encourage avoiding this sentence around line 219 to describe agreement between model & seroprevalence unless the paper aims to present this model as superior to others (numerous issues arise – age-based heterogeneity documented in the early PCR swabs, uncertainty in R_0 , false-negative rates of swabs that this model was fit to, and more), and instead refocus attention on the value of that model for this paper which, as I see it, is a way to translate your estimates of R_0 , contact tracing efficacy and other NPIs into estimates of epidemic size. This edit is not necessary but recommended.
- 3) Related to (2), it seems a major contribution of this paper is evidence that contact tracing in one of the most diligently surveilled populations is insufficient to stop the epidemic – can this be interfaced with / integrated into the work of Kucharski et al. "Effectiveness of isolation, testing, ..."? This is optional, but I feel low-hanging fruit to broaden the impact of this article beyond a survey of V_0 and into a more generalizable program evaluation of TTI for COVID (a topic of extreme relevance given the costs of TTI and, especially here in the US, low rates of case ascertainment & patient compliance raising questions about the cost-effectiveness of this intervention).

Wang X, Tan L, Wang X, et al. Comparison of nasopharyngeal and oropharyngeal swabs for SARS-CoV-2 detection in 353 patients received tests with both specimens simultaneously. *Int J Infect Dis.* 2020;94:107-109. doi:10.1016/j.ijid.2020.04.023

Figures:

I found figure S2 extremely interesting and wonder if this can't be included into the main MS – I feel like one or two rows of Figure S2 could be added to Figure 3 for a comprehensive view of these results (if you chose just one, I'd say the most compelling for me was the age structure of antibody decay rates, which are plotted in aggregate histograms in the bottom of figure 3). If you reduce some of the space between subpanels in figure 3, there's a lot of room to add more rows and provide row & column labels for the subpanels (e.g. column 2 is DiaSorin; row 1 is "titer", row 3 is "Decay rates" etc.) – this can pack more info into this figure while also speeding up digestion. Related: it may be more revealing to show these histograms on the bottom of figure 3 on the same x-scale to reveal the difference and highlight figure 3i as the outlier with a high rate of decay. Finally, since a comparison of different serological assays is a useful contribution of this paper, I might consider color-coding the different plots based on the serological assay and not based on e.g. the month of sampling, the age, etc. (those are more easily seen from the x axis ticks, whereas a common color theme of each test may enable readers to more quickly & intuitively see figure 3 as comparing different tests). One option for the second row of figure 3 is to plot those whose titers rose on a separate subplot vertically aligned to a similar plot of those whose titers fell, and consider if the first and second rows are both necessary or if just one suffices.

Figure 4: I really like these figures. I wonder if it might also be useful to translate subplot b into an additional figure of the expected number of secondary infections by household size. For example, there's a similar secondary attack rate for households containing 2-4 people, but if all else were equal we might then expect twice as many infections to arise in households with 4 people. This isn't mandatory, but if the authors feel this is a compelling addition then I'd agree!

Figure 5: The bar + whisker plots seem like an underutilization of the complex mathematical machinery & output I suspect the authors have from these simulations – points + whiskers can do to save ink and a logit-y scale becomes available if you no longer plot bars from 0 to the expected value. Related the variable R_0 across population, might we also be interested in how the impact of

contact tracing depends on the value of R_0 (related to the Kucharski et al. paper mentioned above)? If you have the data frame of MCMC randomly drawn parameter values and their resultant epidemic sizes, I feel it could be extremely valuable to add an additional plot that generalizes beyond V_0 by illustrating how the final epidemic size estimates (e.g. relative to an unmitigated epidemic) vary as a function of parameters known to vary regionally (TTI recruitment rates, R_0 , etc.). Just another optional thought to maximize the information content & regional generalizability of this figure.

Overall, cool paper – great work to all on your surveillance, management & modelling of the COVID epidemic in this important study population, one of the finer case studies in the world's many regional (& cruise-ship) COVID epidemics. I look forward to seeing this in print.

Best,
Alex

Reviewer #1 (Remarks to the Author):

Thank you for the opportunity to review “SARS-CoV-2 antibody dynamics, within-household transmission and the impact of contact tracing from community-wide serological testing in the Italian municipality of Vo.” The authors should be commended for such deep analyses with these unique data from Vo, Italy. The authors have combined three fairly distinct analyses into one paper: (1) analyses related to the persistence and concordance of different immunoassays, (2) household modeling analyses to estimate the susceptible infectious transmission probability and the cumulative inter-household transmission risk and (3) an modeling exercise to explore the impact of contact tracing of the 100 PCR-confirmed cases in Vo. All three of these analyses are interesting and relevant but combining them all together made this paper really challenging to read (for this humble reviewer at least). The methods, supplement and results do a lot of cross referencing that I was only able to really grasp what the authors were doing when using 2-3 viewing panes at the same time.

We would like to thank the reviewer for the positive feedback and their appreciation of the amount of the work that went into this paper!

The data collected in Vo’ provide the unique opportunity to answer several open questions on SARS-CoV-2 immunity, transmission, and control and we chose to present all these analyses in a single paper to provide a full, unifying picture of the results obtained from the detailed surveillance conducted during the first wave. We appreciate that we provided lots of information and a range of different analyses and thank the reviewer for highlighting that readability can be improved. While cross-referencing is to some extent inevitable, we have limited its use to improve readability.

Summary of changes:

- In the Methods, we have removed the “Household transmission model” section and moved the entire presentation of the household transmission model to the Supplementary Methods, section 4.
- In the Results section, lines 195-196 we have referred to Supplementary Methods, section 4 and clarified that we provide a summary of the methods proposed by Fraser et al. so that the reader does not necessarily have to read the original paper to understand the methods: “Using the model proposed by Fraser et al.²⁴ (summarised in **Supplementary Methods, section 4**) we found that [...]”
- In the Results section, we moved large part of the association analysis results to the Supplementary (lines 163-167, 173-175 and 181-186).
- In the Supplementary Methods, section 5 we have added all details on the transmission model and contact tracing analysis so that the paper is self-contained, and does not rely on the analysis conducted in Lavezzo et al.

Antibody Persistence and Seroprevalence:

- In Figure 2, it appears that the GT baseline assumption results align best with the yellow ‘mean seroprev estimate.’ Is this yellow estimate using the GT baseline assumption as part of the sensitivity estimation? If so, seems a bit circular? If not, would be helpful to clarify exactly what goes into the yellow.

No, the yellow estimate was obtained from the likelihood model described in the Supplementary Methods, Section 3 and is completely independent from the GT definitions. The purpose of this

analysis was indeed to show that the likelihood-based and the baseline GT-based seroprevalence estimates, which were obtained completely independently, compare well.

We agree that the original text was not completely clear, and we have now re-organised the results and methods section to better clarify the difference between the two estimates.

Summary of changes:

- In the Results, lines 120 – 126 we reformulated the presentation of the results as follows: “The estimated SARS-CoV-2 infection prevalence obtained from the multinomial likelihood (yellow horizontal line and shading in **Figure 2**) closely matches the combined assay-specific seroprevalence estimates adjusted by the assay-specific sensitivity and specificity estimated from the model and reported in **Table 5** (panel a, **Figure 2**). The likelihood-based estimate is completely independent from the ground truth definition; however, it closely matches the seroprevalence estimate obtained using the baseline ground truth definition (panel b, **Figure 2**).”
- In the Results section, line 131, we reiterated the independence of the likelihood- and GT-based estimates: “The likelihood method, and the independent utilization of the baseline ground truth definition, overcome the uncertainties associated with the use of the raw seroprevalence estimates [...]”
- We swapped the order of panels a and b in the original Figure 2, so that the GT-based estimates appear next to each other. The new order of panels a and b matches with the order in which we comment on the results in the revised text.
- In the Methods, lines 665-680, we have divided the **Seroprevalence estimates** section into two subsections, namely *Likelihood-based seroprevalence estimate* and *Ground truth-based seroprevalence estimate*. The **Seroprevalence estimates** section now reads as follows: “**Seroprevalence estimates.** Denote the sensitivity and specificity as se and sp , respectively. Denote the raw seroprevalence as $serop_{raw}$, defined as the proportion of samples testing positive (number of positive/number tested). We calculate the adjusted seroprevalence $serop_{adj}$ using formula

$$serop_{adj} = \frac{serop_{raw} + sp - 1}{se + sp - 1} \quad (1)$$

where $se + sp > 1$.

Likelihood-based seroprevalence estimate. We adjust the observed raw seroprevalence with respect to the assay-specific sensitivity and specificity estimated from in-house validation experiments only and using the assay-specific sensitivity and specificity estimated from the model combining the validation experiment and serosurvey results. To infer the true prevalence of infection, we assumed that data from the in-house validation experiments came from a binomial distribution and that the observed results (in terms of case counts in each of the possible assay results categories, **Table 2**) came from a multinomial distribution, having assumed assay independence. Full details on the likelihood formulation and methods used for statistical inference are provided in the **Supplementary Methods, section 3**.

Ground truth-based seroprevalence estimate. In a separate analysis, we adjust the observed raw seroprevalence with respect to the ground truth definition-specific sensitivity and specificity (**Supplementary Table S1**).”

- The Abbott test is not a quantitative test yet the authors seem to use the readouts as precise numeric values that are comparable between batches. I could see using a standard curve to translate this into relative antibody units but I do not think the authors did this. I suggest

motivating this strong assumption or limiting analyses based on numeric readouts to only the quantitative tests. It should also be noted that the Roche N assay has a truncated upper end of the dynamic range which you may want to consider (e.g., see figure in this recent pre-print <https://www.medrxiv.org/content/10.1101/2021.03.16.21253710v1>).

Indeed, both Abbott and Roche tests are not quantitative. Our assumption builds on previous separate analyses used to validate these methods (see Padoan et al, <https://www.sciencedirect.com/science/article/pii/S2352396420304771>).

In Padoan et al. we evaluated the repeatability on five different technical replicates of the same samples performed on four different days and using different batches. We demonstrated that repeatability and within-laboratory precision were in accordance with the repeatability and intermediate precision conditions specified in the international vocabulary of metrology (International vocabulary of metrology (VIM), JCGM 100:2012, https://www.bipm.org/documents/20126/2071204/JCGM_200_2012.pdf/f0e1ad45-d337-bbeb-53a6-15fe649d0ff1?version=1.8&download=true). We had already cited Padoan et al. but we have now added a more precise comment on to justify our assumptions and the use of the Abbott and Roche readouts as quantitative results.

Regarding the truncated upper end of the dynamic range of the Roche assay, we have not observed this issue on our data, as shown in the antibody value distribution below.

Summary of changes:

- In the Methods section, lines 645-647 we added the following sentence:
“Although the Abbott and Roche tests are not quantitative according to the manufacturers’ specifications, we previously demonstrated the repeatability of both methods over different days and using different batches.”

- The authors to a lot of statistical tests on this fairly small dataset to look for differences by age, sex, BMI etc in antibody levels and decay. I suspect that accounting for multiple testing will lead to a fair number of these ‘significant’ differences going away. Also, I think this section of association analyses could be shortened/simplified. The power to detect differences in most of these analyses is super low and calling out each time a difference wasn’t detected might not be necessary if its in a table/plot somewhere.

Thank you for the suggestion. We considered multiple testing before and decided not to adjust for it to facilitate direct comparisons with other publications where only comparison-wise error rate (CER) has been controlled for (Adjusting for multiple testing—when and how? - ScienceDirect). Adjustment

for multiple testing would increase p-values and such adjustments can be made by readers in light of the number of comparisons they wish to consider, because the unadjusted results are correctly reported. In this regard, we have added a comment in the discussion to clarify that we did not account for multiple testing. Moreover, we have significantly shortened and simplified this section, as suggested.

Summary of changes:

- In the Association analysis section, we removed lines 163-167, 173-175 and 181-186.
- In the Discussion section, lines 265-269, we have edited the text as follows:
“Our results, which are presented without accounting for multiple testing to facilitate direct comparisons with other publications where only comparison-wise error rate (CER) has been controlled for, show significant differences in the magnitude of the antibody response by age group and BMI category using all assays, but not by the presence or absence of symptoms.”
- In the Supplementary Material, we added a Supplementary Results section on the association analysis presenting the non-significant tests conducted. It reads as follows:
“Using the baseline ground truth definition, we found no significant trend in the antibody titres with the number of days since symptom onset, and no significant differences in the mean antibody titres of symptomatic versus asymptomatic infections, of hospitalised versus non-hospitalised infections (except for the DiaSorin assay in November, $p = 0.04$), or by sex (**Supplementary Table S3**).
We found no statistically significant difference in the antibody decay rates of symptomatic versus asymptomatic subjects, nor in hospitalized versus non-hospitalized infections, or by sex (except for DiaSorin, $p = 0.05$; **Supplementary Table S4**). Among asymptomatic infections, we observed no significant association between antibody decay rate and BMI (**Supplementary Table S4**). No significant association was found between symptom occurrence and age, nor between symptom occurrence and BMI category, whether or not age groups were included in the model. We also found no significant association between symptoms occurrence and comorbidities (**Supplementary Table S5**) nor between symptoms occurrence and medical treatment (**Supplementary Table S6**).”

- Lines 146-148 – I don’t see any methods for how half-life was estimated.

Good point, we have now added details on the half-life estimation in the Methods at lines 563-565.

Summary of changes:

- In the Methods section, lines 661-664 we have added the following section:
“**Antibody decay rate and half-life estimates.** We estimate the individual-level antibody decay rate as the change in antibody titres observed between May and November (within the same subject) divided by the number of days between the two serosurveys (212 days). The antibody half-life was estimated as the natural logarithm of 0.5 divided by the antibody decay rate.”

- It’s not totally clear to me what data from this study are being used within the Lloyd-Smith model. It would be helpful to clarify how this is connected to the data in this study.

We summarised the relevant equations of the model developed by Lloyd-Smith et al. which were used in our study in lines 169-179 of the Supplementary Material. The results shown in panel c of Figure 5 build on Fraser et al.’s proof that parameter k in the moment generating function of the

distribution of household-size dependent hazards (which we estimated from the fit of the within-household model to the serological data) is equivalent to the shape parameter of the individual reproduction number distribution $f_v(x)$. In other words, panel c in Figure 5 was obtained assuming $R_0 = 2.4$ and by sampling the shape parameter k from the posterior distribution of model V fitted the observed within-household final sizes. We plotted F_{trans} (calculated as described in SI equation 17) against the cumulative distribution function of the individual reproduction number distribution which is also parameterised in terms of R_0 and k . Script Plot_SITP_overdisp_fit_original_model_script_10.R provides the code for this calculation.

Summary of changes:

- In the Supplementary Material, lines 169-179, we have clarified that we followed Lloyd-Smith et al. and provided the equations used as detailed below:
“Following Lloyd-Smith et al.³, we model the expected proportion of transmission due to infectious individuals with reproduction number $v < x$ as

$$F_{trans}(x) = \frac{1}{R_0} \int_0^x u f_v(u) du \quad (17)$$

*where $f_v(x)$ is the probability density function of the individual reproduction number, which was assumed to follow a gamma distribution. Fraser et al.² demonstrate that parameter k in the moment generating function of the distribution of hazards in households of size n is equivalent to the shape parameter of the individual reproduction number distribution $f_v(x)$ ³. The expected proportion of transmission due to individuals with $v > x$ is $1 - F_{trans}(x)$. The proportion of individuals with $v > x$ is $1 - F_v(x)$, where $F_v(x)$ is the cumulative density function of the individual reproduction number distribution. Panel c of **Figure 5** shows $1 - F_{trans}(x)$ on the y-axis versus $1 - F_v(x)$ on the x-axis, having used $R_0 = 2.4$.”*

Household Transmission:

- Great to see an estimate of SITP here and all the model variants considered!

Thank you for the positive feedback!

- From the table in the appendix it seems like model V is not the one with the lowest DIC as suggested in the main text (line 192) nor are the model variants V the ones with the lowest DICs (In 187). Seems like PVXZ and PVZ have the lowest though the Δ DIC is quite small. This might be a good case for model averaging or simply going with the one with the lowest DIC.

Our sentence in the main text was indeed inaccurate. Model V has a DIC of 15.2 and models PVXZ and PVZ have both a DIC of 15.1. It is true that models PVXZ and PVZ have a lower DIC than model V. However, DIC differences > 4 are required for model selection. We agree that model averaging is useful in forecasting and for predictive purposes. However, our study focused on characterising SARS-CoV-2 transmission to identify the mechanisms driving the observed attack rates. For this reason, and to facilitate accessibility and clarity to an already complex model, we prefer to avoid model averaging. In the text, we have corrected our statement and specified that model V is the most parsimonious among the best models (i.e., those with the lowest DIC).

Summary of changes:

- In the Results section, lines 201-203 we revised the text as follows:
“Figure 5 shows the fit of model V, which is the most parsimonious amongst the selected models (i.e., the models with the lowest DIC), to the observed household, individual and secondary attack rates.”

- Since most users won't know what SITP is, I would give a more interpretable definition when providing the result on line 194.

Good point. We have revised the text to introduce the SITP in the Introduction and highlight the difference with the secondary attack rate in the Discussion.

Summary of changes:

- In the Introduction, lines 83-86 we introduce the definition of SITP as follows:
"We also used information on the serological status of 2,566 household members to estimate the Susceptible Infectious Transmission Probability (SITP), which is the probability of SARS-CoV-2 transmission among household members and is an alternative, more nuanced measure of within-household transmission intensity than the within-household secondary attack rate."
- In the Discussion, lines 288-292 we have included a sentence to discuss the difference between the SITP and secondary attack rate within the household:
"To the best of our knowledge, we provide the first estimate of the SITP for SARS-CoV-2, which represents a more nuanced measure of within-household transmission intensity than the within-household secondary attack rate which, by definition, does not account nor control for multiple introductions from outside the household and can thus potentially overestimate the actual extent of transmission occurring within the household setting."

- Is there are reason you don't mention the Q parameter in the results?

No, there was no reason for not mentioning the Q parameter estimates except for consistency with all other parameter estimates which are presented in Supplementary Table S8. However, we have now added a mention to the Q parameter estimate in the main text.

Summary of changes:

- In the Results section, lines 203-206 we added a reference to the Q estimate as follows:
"Using model V, we estimate an escape probability from sources of infection outside the household of 97.8% (95% CrI 97.0% – 98.2%) (or equivalently, that the probability of getting infected from someone outside the household is 2.2% (95% CrI 1.8% – 3.0%)), a within-household Susceptible Infectious Transmission Probability of 27.3% (95% CrI 19.2% – 34.6%) and that 81.8% (95% CrI 55.9 – 95.2%) of SARS-CoV-2 transmission is attributable to the 20% most infectious individuals (Figure 5)."

Contact Tracing Analysis:

- It isn't clear to me exactly what was done here and whether the conclusions that contact tracing probably made little impact are justified. Perhaps if I read the referenced Lavezzo et al paper, this would clarify but this should not be expected of the average reader. My gut is that this analysis should be might be removed from this paper but if not, I would do a lot more to explain the model, how the data were actually used to fit the model (may be simply through se_{ct}?). With clustering in who is traced and clustering in infections (as suggested by the 80/20 stuff) I would have expected to see at least a larger range of results (in addition to a larger modal effect).

We thank the reviewer for pointing out that our contact tracing analysis was not clear.

In the original submission, we built on the model developed in Lavezzo et al. (which was originally fitted to two prevalence measurements) and explored the impact of contact tracing by removing 44% of the infections from the simulated dynamics assuming no lockdown effect (i.e., no reduction

in transmission due to mass testing and lockdown). We had originally interpreted the limited reduction in the simulated epidemic final size (compared to the unmitigated scenario) as evidence that contact tracing alone had a limited impact on the epidemic dynamics and final size.

In the revised version of the manuscript, we have refined our analysis and calibrated the model with contact tracing to the two prevalence measurements, the observed proportion of traced contacts testing positive to the observed proportion of PCR positive infections being traced. This new analysis allowed to disentangle and quantify the impact of mass testing and lockdown from the impact of contact tracing on the epidemic final size. In turn, we could explore the impact of each intervention considered separately (e.g., mass testing and lockdown in the absence of contact tracing and contact tracing in the absence of mass testing and lockdown) and together under a range of scenarios including enhanced and reduced contact tracing. We calibrated the model and estimated the relative reduction in the epidemic final size assuming three values of R_0 (= 2.1, 2.4 and 2.7).

We have revised the Methods and Results sections and added Supplementary Methods, section 5 to provide all details on the model, including how the contact tracing data were used for model calibration and parameter inference. We have also extended the counterfactual analysis and presented a range of results showing the impact of multiple interventions implemented in isolation and in combination, including the effect of enhanced contact tracing.

Summary of changes:

- In the Results section, lines 214-233, we present the revised results as follows:
“To quantify and disentangle the relative impact of contact tracing from the impact of mass testing and lockdown on the observed epidemic dynamics, we extended the transmission model developed in Lavezzo et al.²² to explicitly include contact tracing (**Supplementary Figure S4**). A detailed description of the transmission model is given in the **Supplementary Methods, section 5**. We fitted the model to the observed prevalence of infection among traced contacts (44 out of 190) and sensitivity of contact tracing (44 out of 100) (**Figure 6**, panels b and c) as well as to the observed prevalence of infection in the study population stratified into asymptomatic, pre-symptomatic and symptomatic subjects (**Figure 6**, panel d). Summary statistics of the posterior distributions of the parameters are given in **Supplementary Table S9**. Our results show that, assuming an initial R_0 of 2.4, mass testing and lockdown had a major effect on epidemic control, reducing the basic reproduction number by an average of 83% (95% CrI 65% – 100%) (**Supplementary Table S9**). This finding is confirmed by the small relative reduction in the epidemic final size compared to the unmitigated epidemic obtained when simulating contact tracing in the absence of mass testing and lockdown (**Figure 6**, panel f). On the other hand, the implementation of contact tracing jointly with mass testing and lockdown enhanced epidemic suppression and additional contact tracing efforts would have further reduced the estimated epidemic final size (**Figure 6**, panel g). In the absence of mass testing and lockdown, our results suggest that epidemic suppression comparable in size to that observed during the first wave would have been obtained with a rate of contact tracing and isolation at least four times that implemented during the first wave (**Figure 6**, panel h).”
- We present the results of the extended counterfactual analysis in Figure 6:

- In the Discussion, lines 299-304 we added:

“Our modelling study suggests that, in the absence of mass testing, substantial increases in contact tracing efforts could suppress the epidemic to a similar extent to what has been obtained in Vo’ with mass testing. From a practical perspective however, the substantial contact tracing efforts deployed in Vo’ at the start of the pandemic suggest that the efficiency of traditional contact tracing, based on people’s recollection and reporting, likely reached its maximum in this context.”
- We provide details on the contact tracing analysis in the Methods section, lines 706-739:

“We extended the transmission model developed in Lavezzo et al.²² (see **Supplementary Methods, section 5** for a summary) to include contact tracing. The flow diagram of the model is given in **Supplementary Figure S4**. Contact tracing was modelled by assuming that susceptible and infected subjects (at any stage during infection) could be detected and isolated at rate ct_S and ct_I , respectively. We allowed for two separate detection rates to reflect the simultaneous occurrence of contact tracing with mass testing, implying a higher detection probability for infected subjects compared to susceptible subjects, as suggested by the data. We assumed that traced subjects were fully isolated, i.e., isolated infected subjects did not transmit the disease onwards and quarantined susceptible subjects were protected against the infection. We assumed that all interventions started on 24th February 2020. The effect of mass testing and the lockdown was modelled through a step change in the reproduction number, with $1 - w$ representing the percent reduction in transmission intensity. We simultaneously fitted the prevalence of infection observed in the February and March surveys, the proportion of traced contacts testing PCR positive and the proportion of PCR positive subjects detected by contact tracing by the end of the epidemic. We fitted the model in a Bayesian framework, using the Metropolis-Hastings Markov Chain Monte Carlo (MCMC) method with uniform prior distributions, assuming an initial reproduction number of either 2.1, 2.4, or 2.7 and an average duration of positivity beyond the duration of the infectious period ($1/\sigma$) equal to 4 days. All details on the transmission model are given in **Supplementary Methods, section 5**. In a counterfactual analysis we explored the impact of

different interventions, implemented in isolation and together, on the epidemic final size. The baseline scenario, which was fitted to data, included mass testing, lockdown and contact tracing (MT + CT). We explored the following counterfactual intervention scenarios: (i) mass testing and lockdown without contact tracing (MT); (ii) contact tracing without mass testing and lockdown (CT); (iii) mass testing, lockdown and enhanced or reduced contact tracing (MT + CT, with the specified contact tracing intensity multiplier); (iv) enhanced contact tracing without mass testing and lockdown (CT with the specified contact tracing intensity multiplier). Details on the parameterisation of the counterfactual scenarios are provided in the **Supplementary Methods, section 5.**"

- We added Supplementary Methods, section 5 to provide full details on the model.
- We included a revised version of the flow diagram in Supplementary Figure S4.
- We included Supplementary Table S9 reporting the parameter estimates from model fitting.

Editorial/Minor Comments:

- Ln 39 - I assume this is the SITP. The way in which it is described is not really accurate in the abstract. I suggest rephrasing to be more precise about what this 27.3% means.

Thank you for the suggestion. We have edited the sentence as detailed below.

Summary of changes:

- In the Abstract, lines 38-42 we edited the text as follows:
"Analysis of the serostatus of the members of 1,118 households indicated a 27.3% (95% CrI 19.2%-34.6%) Susceptible-Infectious Transmission Probability (SITP, the probability of SARS-CoV-2 transmission within the household setting), and that 81.8% (95% CrI 55.9%-95.2%) of transmission could be attributed to 20% of infections."

- Line 59 - should be "from an *infectious* household member"

Edited as suggested, thank you.

- Lines 69-71 - please cite references for the 'disappointing results'

We have added a reference to Ferretti et al, Quantifying SARS-CoV-2 transmission suggests epidemic control with digital contact tracing. *Science* 2021; **368**(6491): eabb6936.

<https://science.sciencemag.org/content/368/6491/eabb6936>.

- Line 113 of supplement: I suggest using cumulative incidence in place of prevalence for θ .

We have more directly defined θ as "the probability of having been infected by SARS-CoV-2".

- Figure 4 - should the final sentence of the caption refer to panels a and b rather than b and c?

The final sentence correctly referred to panels b and c because we had already specified the models used in panel a. However, we have revised the caption and made it consistent across panels.

Summary of changes:

- The revised caption of Figure 5 (Figure 4 in the previous version of the manuscript) is:
"Figure 5: Within-household SARS-CoV-2 transmission estimates. (a) Mean and 95% CrI of the Susceptible-Infectious Transmission Probability estimates by household size obtained with model V (red) and PV (green). (b) Mean and 95% CI of the observed household, individual-level and secondary attack rates compared to the mean and 95% CrI of the estimated attack rates obtained with model V. (c) Mean (dark) and 95% CrI (light) proportion of transmission (y-axis) attributable to the most infectious proportion of infections (x-axis)

obtained with model V; homogeneity in transmission would result in $y=x$. (d) Expected number of secondary infections obtained with model V (mean and 95% CrI), showing large variations by household size despite similar secondary attack rates for households of size 2, 3, and 4 (panel b)."

Reviewer #2 (Remarks to the Author):

Who am I?

My name is Alex Washburne and non-anonymous reviews can add the ‘peer’ in peer review for more respectful discussion and assist the authors in understanding the limitations of their reviewer (there are many!). While I have background ranging from immunology to mathematics, ecology and epidemiology, I know enough about immunology to know that I’m not properly equipped to provide comprehensive reviews on the comparisons of different serological assays, different antigens of different SARS-CoV-2 proteins, and more. Consequently, I’ll focus my review efforts on where I may have the most insight: the statistics, modelling, and placement of this article in the broader contexts to which it’s connected.

Overview

The authors present empirical results from extensive nasopharyngeal PCR and serosurvey surveillance of the small and relatively closed population of Vo, Italy. The authors also monitored close-contacts of confirmed cases. The authors estimate the decay (or, in some cases, amplification) of antibody titers after initial confirmed infections and explain variation in waning immunity based on age, BMI, and symptoms of patients. The authors also explain variation in secondary attack rate with household size. Finally, the authors combine these empirical results into estimates of the impact of contact tracing on epidemic suppression. Overall, I feel this paper is well-written and a valuable addition to the literature.

Thank you for the positive feedback, open review and useful inputs provided!

My main comment is the need to reconcile the oropharyngeal/nasopharyngeal swab discrepancies between this and Lavazzo et al.’s previous description of the Feb/March survey (and possibly contending with variable false-negative rates). As it is, these two papers tell different methods for sample collection that make me question which method was in fact used to swab patients in Vo in Feb/March 2020 (Lavazzo et al. describe oropharyngeal swabs of both tonsillar pillars and the oropharynx, whereas this study describes collection of nasopharyngeal swabs; others have reported high false-negative rates of oropharyngeal swabs that aren’t addressed when interfacing the model with the swab data). Otherwise, I provide comments on some presentations of models & figures that I feel can reduce friction & increase information content of this paper, but these aren’t as necessary as the swab issue above.

Thank you for pointing out this apparent inconsistency, which in fact is only in the words used to refer to the swab testing method and not in the method itself. The whole swab procedure used in this paper is identical to the one used in Lavezzo et al. and consists in sampling patients at both the oropharynx and the nasopharynx. The method is detailed in Lavezzo et. al, as noted by the reviewer. Throughout Lavezzo et al. we referred to the swabs as “nasopharyngeal swab” and we carried on with the same terminology in this paper. However, we agree that the mismatched terminology can generate confusion and for clarity, we replaced all occurrences of “nasopharyngeal swab” with “oro-nasopharyngeal swab” throughout the text and in Figure 1.

Major Comments

Use of cumulative incidence vs. seroprevalence

Relationship between percent of population infected and seroprevalence: The y-axis of figure 2, and text relating to figure 2, discusses the “percent of the population infected” as interchangeable with seroprevalence (or seroprevalence adjusting for false-negative rates of serological assays?).

However, surveys in the UK nurses (using Oxford ImmunoTec's assays) have revealed a large body of patients who are infected but don't seroconvert and are identified by T-cells recognizing SARS-CoV-2 specific antigens. I would encourage integrating language about this limitation throughout, referring to "seroprevalence" instead of "percent of population infected" unless explicit adjustments based on T-cell surveys and failure of mild/asymptomatic patients to seroconvert are conducted. Most Se/Sp specs from ground-truthing of serological assays come from studies of confirmed cases, and this may be biased towards patients with severe outcomes more likely to seroconvert.

The reason for using "percent of the population infected" for the GT-derived estimates is that the GT definitions include PCR-positive subjects. In this sense, these estimates are not strictly obtained by serology. However, we agree that this subtle difference may cause confusion and create friction to the narrative and have edited the text and Figure 2 to refer to "seroprevalence" throughout.

Summary of changes:

- In Figure 2, we have swapped panels a and b, edited the y-axis to "seroprevalence" and revised the caption accordingly.

The authors make what I feel is a useful contribution to this discussion by their association analysis finding no relation between symptom severity and antibody titers (this somewhat rebuts my claims above and is in disagreement with literature - e.g. Jiang et al. "Antibody seroconversion" - showing symptom severity having a significant effect on rates of even IgG seroconversion), but that's tucked away in a supplemental table (see my comments below on figures S2 & figure 3 – I feel like some of these supplemental results may add impact to the manuscript and the authors' foundational studies of Vo if presented in the main MS).

Thank you for the positive feedback. We have added a reference to Jiang et al in the Discussion section and have moved Figure S2 to the main text to give it more relevance, as suggested.

Summary of changes:

- In the Discussion, lines 269-272, we added a reference to the results reported in Jiang et al.: "While other studies reported significantly lower IgG levels in asymptomatic versus symptomatic infections during the acute and early convalescent phase^{16,28} and at the time of virus clearance²⁹, our findings suggest that, over the longer term, early differences in the magnitude of antibody response level off."
- We moved Table 6 to the Supplementary
- We moved Figure S2 in the main text; this figure is now referenced as Figure 4.
- We have updated all references to Tables and Figures accordingly.

Swabs:

If I understand this correctly, the authors treat PCR positive results today as similar to PCR positive results in February/March 2020, despite PCR tests in this paper being reported as nasopharyngeal swabs and PCR tests in the Lavazzo et al. paper using the February/March 2020 data being reported as oropharyngeal swabs. Nasopharyngeal and oropharyngeal swabs have different sensitivities & specificities (see Wang et al. below, though many others have evaluated this and estimates vary with many other sources of variation including whether the nasopharynx is swabbed through just one or both nostrils) and require either (a) amending Lavazzo et al if "oropharyngeal" swabs weren't actually used (though detailed description of swabs of tonsillar pillars and the oropharynx were included there), (b) amending this paper to clarify that

oropharyngeal swabs are used throughout, or (c) considering the sensitivity of attack rate and other estimates to different swab procedures. (the authors report in this paper as collecting nasopharyngeal swabs and cite Lavazzo et al “as reported previously”, but that paper reports collecting oropharyngeal swabs). Lavazzo et al. specifically describe swabbing near the tonsils; given the complicated tissue tropism and variable sensitivity of PCR tests of different tissue types, this is a necessary revision and these issues of swabs, false-negative rates, and more come up again in the model parameterization & evaluation (below).

This comment is related to the previous one reporting potential differences in the swabbing technique between this work and the previous one published in *Nature*. As previously clarified, there are no differences in the method used and the wording has been fixed in the revised version of the manuscript.

Model:

The corroboration of seroprevalence estimates with the model in Lavazzo et al. (line 219) raised a few questions for me. First, I noticed that Lavazzo et al. did not include age structure despite an estimated 3-fold increase of PCR positivity in elderly patients reported in that paper – this evidence of major heterogeneity in transmission in Vo raises questions about a homogeneous models’ accuracy in estimating epidemic size, something the authors acknowledge in trying to explain the difference between seroprevalence and model-estimated epidemic sizes but that raises the question of why they didn’t subsequently adjust their estimates of epidemic size to account for this 30% increase in epidemic size in their model vs. seroprevalence estimates (3.5% seroprevalence to 4.9% model-estimated epidemic size).

In Lavezzo et al. we did not include age-structure into the model because the focus of the study was on the natural history of the disease, specifically between asymptomatic, pre-symptomatic and symptomatic infections. It is an interesting idea to extend the dynamical model to include age-structure but given the relatively small sample sizes, we would very likely encounter identifiability issues. While we are keen to explore this in future work, we do not believe this additional analysis would benefit the current paper, which already contains several results. Given that our sentence on the goodness of the estimates obtained in Lavezzo et al. raised this question, we have removed it.

Summary of changes:

- In the Discussion, lines 251-254 we have removed the following sentence:
“[...] and with the expected epidemic final size (mean 4.9%, range 2.9%-8.1%) predicted from a dynamical model previously fitted to the two prevalence estimates observed in February and March 2020²¹, considering that the homogeneous mixing assumption is known to produce larger epidemic final sizes compared to heterogeneous mixing²⁴”.

Second, Lavazzo et al. appear to have fit their model by assuming a zero false-negative rate for the oropharyngeal swabs used – this is not consistent with other studies documenting a sometimes very large (as high as 73%) and provider-dependent false-negative rates (see Wang et al. below; there are many studies investigating false-negative rates of swabs of different types, all of them finding a non-negligible false-negative rate).

It is correct to say that in Lavezzo et al. we assumed perfect sensitivity and specificity for the oronasopharyngeal swabs tests. At the beginning of the pandemic, the PCR test used in our laboratory was validated against the PCR test performed at the National Institute of Health (Istituto Superiore di Sanita’) on PCR positive samples giving 100% agreement. Unfortunately, we do not have quantitative estimates of the sensitivity and specificity of the PCR test used in our laboratory.

However, the agreement observed between the PCR results and the results of the serological tests (Table 4) provides evidence that PCR was indeed highly sensitive and specific. All PCR positive subjects had at least one positive antibody test results and 99.3% of the PCR negative subjects tested negative to at least one serological assay. Of the 14 (0.7%) subjects testing negative to PCR in February/March, 4 reported symptoms before mid-February; these are likely early infections who cleared the virus by the first survey. Of the remaining 10 subjects, 5 were asymptomatic and 5 reported symptoms between 15th February and 20th March. In Lavezzo et al. we explicitly commented on the fact that the model was fitted “assuming perfect diagnostic sensitivity and specificity” and while we cannot exclude that PCR was not 100% accurate, the serological results collected in this study provide further evidence that our assumptions were reasonable.

Third, the authors claim a “good” agreement between Lavazzo et al. estimates and their current seroprevalence, but a midpoint estimate of 4.9% prevalence for Lavazzo et al compared to a 3.5% seroprevalence estimate is not particularly good – the 4.9% midpoint estimate from Lavazzo et al. is outside the 95% credible interval for the serosurvey, so these are different, and that difference raises questions about deeper model-structural uncertainties in the output epidemic sizes from counterfactual simulations (if you change the model structure to account for documented heterogeneity in attack rates by age and that changes epidemic sizes by 30%, then which model structure’s output should we rely on?).

We considered the estimate obtained with the homogeneous mixing model “good” because those estimates are comparable with the estimate obtained in this study (the CIs overlap). This judgment takes into consideration the limited amount of data (2 prevalence estimates) used for calibration and the fact that homogeneous mixing models are known to provide higher attack rates than heterogeneous mixing ones. However, we appreciate that the expected seroprevalence estimates obtained in Lavezzo et al. have limitations and may not be universally perceived as good. As mentioned above, we have removed our sentence comparing the estimate obtained in the current paper with those obtained in Lavezzo et al. to avoid introducing distractions from the main results of the paper.

Fourth, the authors use an R0 estimate of 2.4, but there is considerable regional variation of R0 and others estimate higher R0 (e.g. Sanche et al. estimate an R0 as high as 5.5, which appears consistent with the 2-day doubling times in many densely populated metro areas in the US under generation time distributions reported in the literature) – was there an effort to estimate R0 for Vo? Or, should we evaluate how Vo’s intensive contact tracing would play out across regions with variable R0?

Yes, there is indeed considerable regional variation in R_0 (see for instance Table 1 in Riccardo et al., <https://www.medrxiv.org/content/10.1101/2020.04.08.20056861v1.full.pdf>) and we tried to estimate R_0 from the prevalence data collected in Vo’. However, in the absence of time series data, and to avoid identifiability issues, we opted for exploring plausible scenarios ($R_0 = 2.1, 2.4, 2.7$) which were in line with the observed doubling time (3-4 days in Veneto) and the assumed and estimated serial interval distribution.

It is a good idea to evaluate how the impact of contact tracing on the epidemic final size varies with the assumed value of R_0 . In the revised version of Figure 6 we show the relative reduction in the estimated epidemic final size (compared to the unmitigated scenario) obtained with different interventions under three R_0 scenarios ($R_0 = 2.1, 2.4, 2.7$).

Summary of changes:

- In panels f, g and h of Figure 6, we report the relative reductions in epidemic final size obtained with each intervention for three R_0 scenarios ($R_0 = 2.1, 2.4, 2.7$):

Ultimately, all models are wrong, some are useful, and it seems to me the value of the model for this paper is as a means to estimate final epidemic size under various NPIs. For that end, the model is sufficiently useful and I feel the questions raised above can be avoided by tightening up the treatment of different swab types and not resting as heavily on the model's predictions as such claims raise these more sharp-penciled questions about model quality & prediction accuracy that I feel detract from a sufficiently interesting and impactful empirical paper.

Thank you for clarifying this – we appreciate the concerns and have edited the text to keep the focus on the main and most important messages of the paper.

One could also consider how the model's predicted relative epidemic sizes under various NPIs (relative to the unmitigated scenario) differs from simpler $1-1/((1-a)*R_0)$ calculations where a is the proportion of contacts traced & isolated.

Thank you for the suggestion. Expressing the herd immunity threshold as a function of the proportion of contacts traced and isolated requires assuming that contact tracing and isolation acts only on the susceptible pool (as it would be in standard conditions). In Vo' contact tracing occurred at the same time as mass testing, meaning that contact tracing also detected and isolated exposed contacts during all stages of infection. In this case, we estimated the predicted relative epidemic final sizes numerically.

We revised the contact tracing analysis and calibrated the transmission model (which includes an explicit representation of contact tracing through quarantine/isolation and mass testing/lockdown through a step reduction in R_0) to the observed prevalence of infection among pre-symptomatic, asymptomatic and symptomatic subjects, observed prevalence of infection among traced contacts

and observed proportion of PCR positive subjects detected by contact tracing. This model allowed us to disentangle and quantify the impact of contact tracing from mass testing and lockdown on the observed epidemic dynamics. In a counterfactual analysis, we explored how different NPIs, including larger and smaller proportions of contacts traced and isolated, affect the epidemic final size. We show the results of this analysis in panels e-h of Figure 6.

For more details on the modelling and calibration to the contact tracing data please see our response to reviewer 1's question on Contact Tracing Analysis.

Summary of changes:

- We have shown the dependency of the relative reduction in the epidemic final size on NPIs and contact tracing effort (both in presence and absence of mass testing and lockdown) in Figure 6, panels f-g.
- We explained the methods associated with these results in the Methods, Impact of contact tracing section (lines 702-739) and in the Supplementary Material, section 5.

I raise these issues primarily to provide the authors an n=1 focus group of how the model is received. To alleviate these points of friction, I recommend the following:

1) clarify swab collection and, if different, account for the different false negative rates of different swab types over time. This is the only change I feel is truly necessary for reproducibility.

We have addressed this – it was a presentational issue.

2) It was the qualifier that the model was a “good” fit to seroprevalence that sent me down this hole of evaluating the model’s assumptions & raising sharp-penciled questions about prediction accuracy... I encourage avoiding this sentence around line 219 to describe agreement between model & seroprevalence unless the paper aims to present this model as superior to others (numerous issues arise – age-based heterogeneity documented in the early PCR swabs, uncertainty in R_0 , false-negative rates of swabs that this model was fit to, and more), and instead refocus attention on the value of that model for this paper which, as I see it, is a way to translate your estimates of R_0 , contact tracing efficacy and other NPIs into estimates of epidemic size. This edit is not necessary but recommended.

Agreed – we have removed the sentence.

3) Related to (2), it seems a major contribution of this paper is evidence that contact tracing in one of the most diligently surveilled populations is insufficient to stop the epidemic – can this be interfaced with / integrated into the work of Kucharski et al. “Effectiveness of isolation, testing, ...”? This is optional, but I feel low-hanging fruit to broaden the impact of this article beyond a survey of V_0 and into a more generalizable program evaluation of TTI for COVID (a topic of extreme relevance given the costs of TTI and, especially here in the US, low rates of case ascertainment & patient compliance raising questions about the cost-effectiveness of this intervention).

Thank you for the suggestion and reference. We have extended the analysis to explore the extent to which increased contact tracing efforts in the absence of mass testing and lockdown would have produced similar epidemic final sizes to the observed one. We find that the rates of isolation should have been at least 4 times larger than those put in place at the beginning of the pandemic to obtain similar levels of epidemic suppression to those observed in V_0' during the first wave. We have presented these new results in panel h of Figure 6.

Summary of changes:

- We have revised Figure 6 and included panel h showing the impact of enhanced contact tracing efforts in the absence of mass testing and lockdown.

Wang X, Tan L, Wang X, et al. Comparison of nasopharyngeal and oropharyngeal swabs for SARS-CoV-2 detection in 353 patients received tests with both specimens simultaneously. Int J Infect Dis. 2020;94:107-109. doi:10.1016/j.ijid.2020.04.023

Figures:

I found figure S2 extremely interesting and wonder if this can't be included into the main MS – I feel like one or two rows of Figure S2 could be added to Figure 3 for a comprehensive view of these results (if you chose just one, I'd say the most compelling for me was the age structure of antibody decay rates, which are plotted in aggregate histograms in the bottom of figure 3).

We removed Table 6 and included Supplementary Figure S2 in the main text, which appears as Figure 4 in the revised version of the manuscript.

Summary of changes:

- We have removed Table 6 from the main text and included it in the Supplementary Materials as Supplementary Table S7.
- We have moved Supplementary Figure S2 into the main text as Figure 4 in the revised manuscript.

If you reduce some of the space between subpanels in figure 3, there's a lot of room to add more rows and provide row & column labels for the subpanels (e.g. column 2 is DiaSorin; row 1 is "titer", row 3 is "Decay rates" etc.) – this can pack more info into this figure while also speeding up digestion. Related: it may be more revealing to show these histograms on the bottom of figure 3 on the same x-scale to reveal the difference and highlight figure 3i as the outlier with a high rate of decay. Finally, since a comparison of different serological assays is a useful contribution of this paper, I might consider color-coding the different plots based on the serological assay and not based on e.g. the month of sampling, the age, etc. (those are more easily seen from the x axis ticks, whereas a common color theme of each test may enable readers to more quickly & intuitively see figure 3 as comparing different tests). One option for the second row of figure 3 is to plot those whose titers rose on a separate subplot vertically aligned to a similar plot of those whose titers fell, and consider if the first and second rows are both necessary or if just one suffices.

Thank you for these suggestions, we modified: (i) color codes, (ii) titles of rows and columns, (iii) split the individual-level dynamics into increasing and decreasing changes, (iv) adjusted the scales of the decay rates and (v) divided the positive and negative trends in two separate rows. We would prefer to keep the first two rows of Figure 3 because the information provided is different.

Summary of changes:

- Figure 3 has been edited as suggested and the revised figure is shown below:

Figure 4: I really like these figures. I wonder if it might also be useful to translate subplot b into an additional figure of the expected number of secondary infections by household size. For example, there's a similar secondary attack rate for households containing 2-4 people, but if all else were equal we might then expect twice as many infections to arise in households with 4 people. This isn't mandatory, but if the authors feel this is a compelling addition then I'd agree!

Yes, the intuition is correct. We have added panel d to Figure 5 (Figure 4 in the original submission) to show the expected number of non-primary infections by household size.

Summary of changes:

- We have added panel d to Figure 5 (originally Figure 4) showing the expected number of non-primary infections by household size:

Figure 5: The bar + whisker plots seem like an underutilization of the complex mathematical machinery & output I suspect the authors have from these simulations – points + whiskers can do to save ink and a logit-y scale becomes available if you no longer plot bars from 0 to the expected value. Related the variable R_0 across population, might we also be interested in how the impact of contact tracing depends on the value of R_0 (related to the Kucharski et al. paper mentioned above)? If you have the data frame of MCMC randomly drawn parameter values and their resultant epidemic sizes, I feel it could be extremely valuable to add an additional plot that generalizes beyond V_0 by illustrating how the final epidemic size estimates (e.g. relative to an unmitigated epidemic) vary as a function of parameters known to vary regionally (TTI recruitment rates, R_0 , etc.). Just another optional thought to maximize the information content & regional generalizability of this figure.

Thank you for the inputs and thoughts. We have revised Figure 6 (originally Figure 5) as suggested. Please see our reply to comment 3) above.

Overall, cool paper – great work to all on your surveillance, management & modelling of the COVID epidemic in this important study population, one of the finer case studies in the world's many regional (& cruise-ship) COVID epidemics. I look forward to seeing this in print.

**Best,
Alex**

P.S. If the authors have any questions that they feel can expedite the next round of revisions if discussed back & forth, they are free to contact me at alex.d.washburne@gmail.com

Thank you for reviewing our paper and all comments and suggestions.

REVIEWERS' COMMENTS

Reviewer #1 (Remarks to the Author):

Great job on the revisions. My only remaining minor comment is to please be clear about the interpretation of the SITP parameter. It is not simply "the probability of SARS-CoV-2 transmission among household members." I suggest referring to Fraser et al (2011, 10.1093/aje/kwr122), which included co-authors of this paper for a more clear interpretation.

Reviewer #2 (Remarks to the Author):

The authors have done a magnificent job incorporating my feedback and addressing the few issues I raised, most importantly on the swab methodology. Fantastic work, and I can't wait to see what we learn next!

Best
Alex Washburne

Point-by-point response letter

SARS-CoV-2 antibody dynamics and transmission from community-wide serological testing in the Italian municipality of Vo'

Ilaria Dorigatti, Enrico Lavezzo, Laura Manuto, Constanze Ciavarella, Monia Pacenti, Caterina Boldrin, Margherita Cattai, Francesca Saluzzo, Elisa Franchin, Claudia Del Vecchio, Federico Caldart, Gioele Castelli, Michele Nicoletti, Eleonora Nieddu, Elisa Salvadoretti, Beatrice Labella, Ludovico Fava, Simone Guglielmo, Mariateresa Fascina, Marco Grazioli, Gualtiero Alvisi, Maria Cristina Vanuzzo, Tiziano Zupo, Reginetta Calandrin, Vittoria Lisi, Lucia Rossi, Ignazio Castagliuolo, Stefano Merigliano, H. Juliette T. Unwin, Mario Plebani, Andrea Padoan, Alessandra R. Brazzale, Stefano Toppo, Neil M. Ferguson, Christl A. Donnelly, Andrea Crisanti

Reviewer #1 (Remarks to the Author):

Great job on the revisions. My only remaining minor comment is to please be clear about the interpretation of the SITP parameter. It is not simply "the probability of SARS-CoV-2 transmission among household members." I suggest referring to Fraser et al (2011, 10.1093/aje/kwr122), which included co-authors of this paper for a more clear interpretation.

Good point, thank you for bringing this to our attention. Due to editorial requests we had to remove the short description that was given in the abstract. However, we provide the exact definition of the SITP, as given in Fraser et al., in lines 86 – 91.

Summary of changes:

- In lines 86 – 91 we define the SITP as follow:
"We also used information on the serological status of 2,566 household members to estimate the Susceptible Infectious Transmission Probability (SITP), which is the probability of SARS-CoV-2 transmission occurring between each susceptible-infectious pair of individuals, measured over the whole period of infectiousness of the infectious individual and in the case when the susceptible individual is not infected by a third party during that period, and is an alternative, more nuanced measure of within-household transmission intensity than the within-household secondary attack rate."

Moreover, during the final revisions of the household-transmission model results, we decided to increase the size of the sampled parameter realisations from 100 to 1,000. This makes the estimates of the SITP and overdispersion less variable and dependent on the sampled parameter realisations. Using 1,000 samples the SITP is 26.0% (95% CrI 17.2% – 36.9%) and we estimate that 79.1% (95% CrI 48.9 – 98.9%) of SARS-CoV-2 transmission is attributable to the 20% most infectious individuals. The previous estimates were 27.3% (95% CrI 19.2% – 34.6%) and 81.8% (95% CrI 55.9 – 95.2%), respectively. We have edited the text accordingly.

Summary of changes:

- In the Abstract, lines 37-38 we updated the text as follow:
"Analysis of the serostatus of the members of 1,118 households indicates a 26.0% (95% CrI 17.2%-36.9%) Susceptible-Infectious Transmission Probability."
- In the Results section, lines 183-188 we report the updated estimates:
"Using model V, we estimate an escape probability from sources of infection outside the household of 97.8% (95% CrI 97.0% – 98.2%) (or equivalently, that the probability of getting infected from someone outside the household is 2.2% (95% CrI 1.8% – 3.0%)), a within-

household Susceptible Infectious Transmission Probability (SITP) of 26.0% (95% CrI 17.2% – 36.9%) and that 79.1% (95% CrI 48.9 – 98.9%) of SARS-CoV-2 transmission is attributable to the 20% most infectious individuals (Figure 5)."

Reviewer #2 (Remarks to the Author):

The authors have done a magnificent job incorporating my feedback and addressing the few issues I raised, most importantly on the swab methodology. Fantastic work, and I can't wait to see what we learn next!

**Best
Alex Washburne**

Thank you!